# Learning to Play Multi-Follower Bayesian Stackelberg Games

**Gerson Personnat, Tao Lin, Safwan Hossain, David C. Parkes**[*]
John A. Paulson School of Engineering and Applied Sciences
Harvard University

## Abstract

In a multi-follower Bayesian Stackelberg game, a leader plays a mixed strategy over $L$ actions to which $n \geq 1$ followers, each having one of $K$ possible private types, best respond. The leader's optimal strategy depends on the distribution of the followers' private types. We study an online learning version of this problem: a leader interacts for $T$ rounds with $n$ followers with types sampled from an unknown distribution every round. The leader's goal is to minimize regret, defined as the difference between the cumulative utility of the optimal strategy and that of the actually chosen strategies. We design learning algorithms for the leader under different feedback settings. Under type feedback, where the leader observes the followers' types after each round, we design algorithms that achieve $\mathcal{O}\big(\sqrt{\min\{L\log(nKAT),\ nK\} \cdot T}\big)$ regret for independent type distributions and $\mathcal{O}\big(\sqrt{\min\{L\log(nKAT),\ K^n\} \cdot T}\big)$ regret for general type distributions. Interestingly, those bounds do not grow with $n$ at a polynomial rate. Under action feedback, where the leader only observes the followers' actions, we design algorithms with $\mathcal{O}(\min\{\sqrt{n^L K^L A^{2L} LT \log T},\ K^n \sqrt{T} \log T\})$ regret. We also provide a lower bound of $\Omega(\sqrt{\min\{L,\ nK\}T})$, almost matching the type-feedback upper bounds.

## 1 Introduction

Stackelberg games are a fundamental model of strategic interaction in multi-agent systems. Unlike normal-form games where all agents simultaneously play their strategy, Stackelberg games model a *leader* committing to their strategy; the remaining *follower(s)* take their actions after observing the leader's commitment (Conitzer & Sandholm, 2006; Von Stackelberg, 2010). Such asymmetric interactions are ubiquitous in a wide range of setting, from a firm entering a market dominated by an established competitor (Von Stackelberg, 2010), to an online platform releasing features that influence consumers on that platform (Zhao et al., 2023; Cao et al., 2024), to security games (Balcan et al., 2015; Sinha et al., 2018) to strategic machine learning (Hardt et al., 2016; Hossain et al., 2025). They also form the foundation of seminal models in computational economics like Bayesian Persuasion (Kamenica & Gentzkow, 2011) or contract design (Dütting et al., 2024) that capture more structured settings with asymmetries relating to information or payouts respectively.

In these settings and beyond, there is one key question: what is the optimal strategy for the leader to commit to? Answering this question requires knowing how the follower(s) will react to the leader's strategy, which typically boils down to knowing the followers' utilities. The Bayesian approach attempts to relax this complete information assumption. Pioneering works like Conitzer & Sandholm (2006) assume that followers' utilities are parametrized by hidden types from a known distribution. Here, the leader aims to compute the Bayesian Stackelberg Equilibrium: the strategy maximizing the leader's expected utility with the followers' types drawn from the known distribution.

In many of the settings mentioned, even the Bayesian perspective may be too strong and unrealistic: the leader (e.g., online platform, dominant firm) may only know the structure of the followers' utilities but not the distribution of their types (Cole & Roughgarden, 2014). While not much can be achieved in a single round of such a game, the leader can often interact with the followers over multiple rounds and learn about them over time. The leader must, however, balance learning with playing the optimal

---

[*]correspondence to `lintao@cuhk.edu.cn` or `shossain@g.harvard.edu`.

strategy given current information – the well-known exploration-exploitation trade-off in the online learning literature.

**Our Contributions:** This paper comprehensively studies the learning and computational problem for an online Bayesian Stackelberg game (BSG). Specifically, we consider the interaction over $T$ rounds between a leader and $n$ followers, each realizing one of $K$ possible private types at each round. To our knowledge, this is the first work on online learning in BSGs with multiple followers. We study two feedback models: observing realized types of the followers, or observing their best-responding actions, after each round. Our core objective is exploring how these feedback models affect the learnability of the optimal strategy, which is challenging for several reasons. First, with multiple followers, the unknown joint type space is exponentially large. Further, followers' taking best-responding actions means that the leader's utility function is discontinuous and non-convex. Lastly, even the offline single-follower version of this problem has known computational challenges (Conitzer & Sandholm, 2006). A key technical tool used to unravel this is a geometric characterization of the leader's strategy space in terms of best-response regions (presented in Section 3). Section 4 uses this and an observation about learning type distributions vis-a-vis learning utility to provide algorithms for both general type distributions and independent ones. A matching lower bound is also provided. Section 5 then studies algorithms for the action feedback case, where we leverage our geometric insights along with techniques from linear bandits. Table 1 summarizes our results. Throughout, we comment on the computational complexity of our algorithms and uncover interesting trade-offs that situate our work with the broader literature on Stackelberg games.

Table 1: Regret bounds for learning the optimal leader strategy in Bayesian Stackelberg games with $n$ followers under various settings. The $\widetilde{\mathcal{O}}(\cdot)$ notation omits logarithmic factors.

| | Type Feedback | | Action Feedback |
|---|---|---|---|
| | **Independent types** | **General types** | |
| **Upp. Bound** | $\widetilde{\mathcal{O}}(\sqrt{\min\{L,\, nK\}T})$ | $\widetilde{\mathcal{O}}(\sqrt{\min\{L,\, K^n\}T})$ | $\widetilde{\mathcal{O}}(\min\{\sqrt{n^L K^L A^{2L} L},\, K^n\}\sqrt{T})$ |
| **Low. Bound** | $\Omega(\sqrt{\min\{L,\, nK\}T})$ | $\Omega(\sqrt{\min\{L,\, nK\}T})$ | $\Omega(\sqrt{\min\{L, nK\}T})$ |

**Related Works:** Our work contributes to the growing literature on the computational and learning aspects of Stackelberg games (Conitzer & Sandholm, 2006; Conitzer & Korzhyk, 2011; Castiglioni et al., 2020; Zhu et al., 2023). In particular, Letchford et al. (2009); Peng et al. (2019); Bacchiocchi et al. (2024) study online learning in non-Bayesian Stackelberg games, while Bollini et al. (2026) consider Bayesian Stackelberg games, both with a single follower. They assume that the follower myopically best responds in each round, which we also assume. However, their goal is to learn the follower's unknown utility function, whereas we aim to learn the unknown type distribution of the followers with known utility functions.

Balcan et al. (2015; 2025) design online learning algorithms with poly$(K)\sqrt{T}$ regrets for Bayesian Stackelberg games with a single follower with unknown type distribution, while we consider multiple followers. Bernasconi et al. (2023) obtain $\widetilde{\mathcal{O}}(K^{3n/2}\sqrt{T})$ regret for multi-receiver Bayesian persuasion problem (which is similar to multi-follower Bayesian Stackelberg game) by a reduction to adversarial linear bandit problem. Adopting Bernasconi et al. (2023)'s technique to our setting would lead to poly$(K^n)\sqrt{T}$ regret, which is exponential in the number of followers $n$ (see details in Section 5) and undesirable when followers are many. Using a different approach, we design an algorithm with $\widetilde{\mathcal{O}}(\sqrt{n^L K^L A^{2L} LT})$ regret, a better result when the number of leader's actions $L$ is small compared to the number of followers $n$. The exponential dependency on $L$ is unavoidable from a computational perspective, as Conitzer & Sandholm (2006) show that BSGs are NP-Hard to solve with respect to $L$. Our algorithm combines the Upper Confidence Bound (UCB) principle and a partition of the leader's strategy space into best-response regions, which is a novel approach to our knowledge.

Online Bayesian Stackelberg game can be seen as a piecewise linear stochastic bandit problem. While linear stochastic bandit problems have been well studied (Auer et al., 2002; Dani et al., 2008; Abbasi-yadkori et al., 2011), piecewise linearity brings additional challenges. Bacchiocchi et al. (2025) study a single-dimensional piecewise linear stochastic bandit problem with unknown pieces; in contrast, we have known pieces but a multi-dimensional space, so the techniques and results of that work and ours are not directly comparable.

## 2 MODEL

**Multi-Follower Bayesian Stackelberg Game:** We consider the interactions between a single *leader* and $n \geq 1$ *followers*. The leader has $L \geq 2$ actions, denoted by $\mathcal{L} = [L] = \{1, \ldots, L\}$, and chooses a mixed strategy $x \in \Delta(\mathcal{L})$ over them, where $\Delta(\mathcal{L})$ is the space of probability distributions over the action set. We use $x(\ell)$ to denote the probability of the leader playing action $\ell \in \mathcal{L}$.[1] Each follower has a finite action set $\mathcal{A} = [A]$. We represent the joint action of the $n$ followers as $\boldsymbol{a} = (a_1, ..., a_n)$. Each follower $i$ also has a private type $\theta_i \in \Theta = [K]$, with the vector of all follower types denoted by $\boldsymbol{\theta} = (\theta_1, ..., \theta_n) \in \Theta^n$. We consider a Bayesian setting where this type vector is drawn from a distribution $\boldsymbol{\mathcal{D}}$ (i.e., $\boldsymbol{\theta} \sim \boldsymbol{\mathcal{D}}$), with $\mathcal{D}_i$ denoting the marginal distribution of $\theta_i$. The properties of this joint distribution play a key role in our results. We will consider two scenarios:

- Independent type distributions: The followers' types are independent: $\boldsymbol{\mathcal{D}} = \mathcal{D}_1 \times \cdots \times \mathcal{D}_n$.
- General type distributions: The followers' types can be arbitrarily correlated.

If the leader selects action $\ell \in \mathcal{L}$ and the followers select joint action $\boldsymbol{a} \in \mathcal{A}^n$, the leader receives utility $u(\ell, \boldsymbol{a}) \in [0, 1]$ and each follower $i$ receives utility $v_i(\ell, a_i, \theta_i) \in [0, 1]$. Observe that each follower's utility depends only on their own action and type, alongside the leader's action; it does not depend on the actions of other followers.[2] For a leader's mixed strategy $x \in \Delta(\mathcal{L})$ and followers' actions $\boldsymbol{a}$, the leader's expected utility is denoted by $u(x, \boldsymbol{a}) = \mathbb{E}_{\ell \sim x}[u(\ell, \boldsymbol{a})] = \sum_{\ell \in \mathcal{L}} x(\ell) u(\ell, \boldsymbol{a})$. Likewise, the $i$th follower's expected utility under $x$ is $v_i(x, a_i, \theta_i) = \mathbb{E}_{\ell \sim x}[v_i(\ell, a_i, \theta)]$. We assume that the leader knows each follower's utility function but not their private types.

An instance of a multi-follower Bayesian Stackelberg game is defined by the tuple $I = (n, L, A, K, u, v, \boldsymbol{\mathcal{D}})$. In this game, the leader first commits to a mixed strategy $x$ without knowledge of the followers' types. The follower types are then jointly realized from $\boldsymbol{\mathcal{D}}$, and each follower selects a *best-responding* action based on the leader's strategy. It is without loss of generality to consider followers choosing pure action since follower utilities are independent of one another.

**Definition 2.1** (Followers' Best Response). *For a mixed strategy $x$ of the leader, the best response of a follower $i$ with realized type $\theta_i$ is given by $\mathrm{br}_i(\theta_i, x) \in \arg\max_{a \in \mathcal{A}} v_i(x, a, \theta_i)$.[3] The vector of best responses is denoted by $\mathbf{br}(\boldsymbol{\theta}, x) = (\mathrm{br}_1(\theta_1, x), ..., \mathrm{br}_n(\theta_n, x))$.*

Let $U_{\boldsymbol{\mathcal{D}}}(x) = \mathbb{E}_{\boldsymbol{\theta} \sim \boldsymbol{\mathcal{D}}}[u(x, \mathbf{br}(\boldsymbol{\theta}, x))]$ denote the leader's expected utility when the leader commits to mixed strategy $x$, the followers have their types drawn from $\boldsymbol{\mathcal{D}}$ and best respond.

**Definition 2.2** (Leader's Optimal Strategy). *For a joint follower type distribution $\boldsymbol{\mathcal{D}}$, an* optimal *strategy for the leader, also known as a* Stackelberg Equilibrium*, is given by:*

$$x^* \in \arg\max_{x \in \Delta(\mathcal{L})} U_{\boldsymbol{\mathcal{D}}}(x) = \arg\max_{x \in \Delta(\mathcal{L})} \mathbb{E}_{\boldsymbol{\theta} \sim \boldsymbol{\mathcal{D}}}[u(x, \mathbf{br}(\boldsymbol{\theta}, x))].$$

**Online Learning Model:** When the leader knows the distribution $\boldsymbol{\mathcal{D}}$, they can compute the optimal strategy by solving the optimization problem specified in Definition 2.2. Indeed, this is the premise of Conitzer & Sandholm (2006). Our work examines an online learning model where the leader does not know the type distribution $\boldsymbol{\mathcal{D}}$ a priori; instead, the leader must learn the optimal strategy through feedback from repeated interactions with followers over $T$ rounds.

We examine two feedback models. In the *type feedback* setting, the leader observes the types $\boldsymbol{\theta}^t$ of the followers after each round $t$, whereas in the *action feedback* setting, the leader only observes the actions $\boldsymbol{a}^t$ of the followers. Note that type feedback is strictly more informative than action feedback since the follower's actions can be inferred from their types by computing their best response (Definition 2.1). We summarize the interactions at a given round $t$ as follows:

1. The leader chooses a strategy $x^t \in \Delta(\mathcal{L})$.
2. Follower types for this round are realized: $\boldsymbol{\theta}^t \sim \boldsymbol{\mathcal{D}}$.

---

[1] All of our results can be generalized to the setting where the leader's strategy space is an arbitrary compact convex set $\mathcal{X} \subseteq \mathbb{R}^d$ and the leader and followers' utility functions $u(x, \boldsymbol{a}), v_i(x, a_i, \theta_i)$ are linear (or affine) functions of $x \in \mathcal{X}$. We focus on the probability simplex $\mathcal{X} = \Delta(\mathcal{L})$ to simplify presentation.

[2] This no externality assumption is common in modeling a large population of agents (Dughmi & Xu, 2017; Xu, 2020; Castiglioni et al., 2020). We discuss this assumption more in Section 6.

[3] In case of ties, we assume that followers break ties in favor of the leader.

3. Followers take their best-responding actions $\boldsymbol{a}^t = \mathbf{br}(\boldsymbol{\theta}, x^t)$ and the leader gets utility $u(x^t, \boldsymbol{a}^t)$.

4. Under type feedback, the leader observes the type profile $\boldsymbol{\theta}^t$. Under action feedback, the leader observes only the followers' actions $\boldsymbol{a}^t$.

The leader deploys a learning algorithm (based on past feedback) to select strategy $x^t$ for every round $t$. We study learning algorithms that minimize the cumulative *regret* with respect to the optimal equilibrium strategy (Definition 2.2). Formally defined below, minimizing this objective requires a careful balance between exploring the strategy space while not taking too many sub-optimal strategies.

**Definition 2.3.** *The* regret *of a learning algorithm that selects strategy $x^t$ at round $t \in [T]$ is:*

$$\mathrm{Reg}(T) = \sum_{t=1}^{T} \mathbb{E}_{\boldsymbol{\theta}^t \sim \boldsymbol{\mathcal{D}}}\Big[u(x^*, \mathbf{br}(\boldsymbol{\theta}^t, x^*)) - u(x^t, \mathbf{br}(\boldsymbol{\theta}^t, x^t))\Big] = \sum_{t=1}^{T} \Big(U_{\boldsymbol{\mathcal{D}}}(x^*) - U_{\boldsymbol{\mathcal{D}}}(x^t)\Big).$$

Note that $\mathrm{Reg}(T)$ is a random variable, because the selection of $x^t$ depends on the past type realizations $\boldsymbol{\theta}^1, ..., \boldsymbol{\theta}^{t-1}$. We aim to minimize the expected regret $\mathbb{E}[\mathrm{Reg}(T)]$.

Lastly, our model assumes followers behave *myopically*, selecting their best actions based only on the leader's current strategy, without considering future rounds. This is consistent with the literature on learning in Stackelberg games (see related works) and well-motivated in settings like online platforms or security games where followers maximize their immediate utility.

## 3 BEST RESPONSE REGIONS: A GEOMETRIC PERSPECTIVE

Since the followers' best-responding actions are sensitive to the leader's strategy $x$, the leader's expected utility function $U_{\boldsymbol{\mathcal{D}}}(x)$ is discontinuous in $x$. This presents a key challenge to both learning and optimizing over the leader's strategy space. To overcome this challenge, we first show that the leader's strategy space $\Delta(\mathcal{L})$ can be partitioned into a polynomial number of non-empty *best-response regions* (followers have the same best-response actions within each region). While the notion of best-response regions has been proposed by prior works (Balcan et al., 2015; Peng et al., 2019; Bacchiocchi et al., 2024; Yang & Zhang, 2024), those works consider single-follower cases. With multiple followers, we will argue that the number of such regions does not increase exponentially in the number of followers $n$ (Lemma 3.2) – a key property to be used in later sections. At a high level, the best-response region approach allows us to reason about the leader strategy space in a discrete sense. This is not only instructive for regret analysis (such as for Theorem 4.1) but also facilitates leveraging algorithms like UCB (Algorithm 3), which are defined for discrete settings.

### 3.1 A SINGLE FOLLOWER

To build intuition, we first consider the leader playing against a single follower ($n = 1$). The follower has a utility function $v(\ell, a, \theta)$ and a type $\theta \in \Theta = [K]$ drawn from distribution $\mathcal{D}$. Next, let $w : \Theta \to \mathcal{A}$ be a mapping from follower type to action – i.e. $w(\theta)$ specifies an action for type $\theta$. For such a mapping $w$, let $R(w) \subseteq \Delta(\mathcal{L})$ be the set of leader strategies under which the follower's best response action $\mathrm{br}(\theta, x)$ is equal to $w(\theta)$ for every type $\theta \in \Theta$. Formally:

$$R(w) = \Big\{x \in \Delta(\mathcal{L}) \,\big|\, \mathrm{br}(\theta, x) = w(\theta), \ \forall \theta \in \Theta\Big\}$$
$$= \Big\{x \in \Delta(\mathcal{L}) \,\big|\, v(x, w(\theta), \theta) \geq v(x, a', \theta), \ \forall \theta \in \Theta, \forall a' \in \mathcal{A}\Big\}$$

where we recall that for any action $a$, $v(x, a, \theta) = \sum_{\ell \in \mathcal{L}} x(\ell) v(\ell, a, \theta)$. The set $R(w)$ is defined as the *best-response region* for mapping $w$. This region can also equivalently be defined as the intersection of several halfspaces (see Figure 1 in Appendix A for a visual). In particular, let

$$d_{\theta, a, a'} = \big[v(1, a, \theta) - v(1, a', \theta) \,,\, \ldots \,,\, v(L, a, \theta) - v(L, a', \theta)\big]^T \in \mathbb{R}^L$$

denote the "advantage" of follower type $\theta$ taking action $a$ over $a'$ at each of the $L$ possible leader actions. Then the halfspace $H(d_{\theta, a, a'}) = \big\{x \in \Delta(\mathcal{L}) \,|\, \langle x, d_{\theta, a, a'}\rangle \geq 0\big\}$ contains all the leader strategies under which the follower with type $\theta$ prefers action $a$ over $a'$. Thus, the best-response region is $R(w) = \bigcap_{\theta \in \Theta, a \in \mathcal{A}} H(d_{\theta, w(\theta), a})$, the intersection of $|\Theta| \cdot |\mathcal{A}| = KA$ halfspaces.

## 3.2 MULTIPLE FOLLOWERS

We generalize the intuitions from the single-follower case to the multi-follower case. Let $W = (w_1, \ldots, w_n)$ denote a tuple of $n$ mappings, where each $w_i : \Theta \to \mathcal{A}$ is the best-response mapping for follower $i$. So, $W$ is a mapping from joint type space $\Theta^n$ to joint action space $\mathcal{A}^n$, where $W(\boldsymbol{\theta}) = (w_1(\theta_1), \ldots, w_n(\theta_n)) \in \mathcal{A}^n$ denotes the joint action of all followers under joint type $\boldsymbol{\theta} = (\theta_1, \ldots, \theta_n)$. Alternatively, one can think of $W$ as a matrix, $W = (w_{ik}) \in \mathcal{A}^{n \times K}$, where each entry $w_{ik} \in \mathcal{A}$ records the best-response action for follower $i$ if he has type $\theta_i = k$. We generalize the notion of best-response region $R(w)$ from the singe-follower case to the multi-follower case:

**Definition 3.1** (Best-Response Region). *For a matrix $W \in \mathcal{A}^{n \times K}$, the best-response region for $W$ is the set of leader strategies under which the followers' best responses are given by $W$:*

$$R(W) = \left\{ x \in \Delta(\mathcal{L}) \mid \mathbf{br}(\boldsymbol{\theta}, x) = W(\boldsymbol{\theta}), \quad \forall \boldsymbol{\theta} \in \Theta^n \right\}.$$

As in the single-follower case, $R(W)$ can be expressed as the intersection of multiple halfspaces: $R(W) = \bigcap_{i \in [n], \theta_i \in \Theta, a_i \in \mathcal{A}} H(d_{\theta_i, w_i(\theta_i), a_i})$.

We make an important observation: the leader's expected utility function $U_{\mathcal{D}}(x)$ is linear in $x$ within each non-empty best-response region. By definition, for all $\boldsymbol{\theta} \in \Theta^n$ and $x \in R(W)$, we have $\mathbf{br}(\boldsymbol{\theta}, x) = W(\boldsymbol{\theta})$. So,

$$U_{\mathcal{D}}(x) = \sum\nolimits_{\boldsymbol{\theta} \in \Theta^n} \mathcal{D}(\boldsymbol{\theta}) u(x, \mathbf{br}(\boldsymbol{\theta}, x)) = \sum\nolimits_{\boldsymbol{\theta} \in \Theta^n} \mathcal{D}(\boldsymbol{\theta}) \sum_{\ell \in \mathcal{L}} x(\ell) u(\ell, W(\boldsymbol{\theta})) = \sum\nolimits_{\boldsymbol{\theta} \in \Theta^n} \mathcal{D}(\boldsymbol{\theta}) \langle x, z_{W, \boldsymbol{\theta}} \rangle.$$

where $z_{W, \boldsymbol{\theta}}$ is the $L$-dimensional vector $z_{W, \boldsymbol{\theta}} = (u(1, W(\boldsymbol{\theta})), ..., u(L, W(\boldsymbol{\theta})))$. So, we conclude that the leader's expected utility is linear within each region $R(W)$:

**Lemma 3.1.** *For each $W$, the leader's expected utility function $U_{\mathcal{D}}(x)$ is linear in $x \in R(W)$.*

Although $U_{\mathcal{D}}(x)$ is linear *within* each best-response region, it could be non-linear and even discontinuous *across* different best-response regions.

## 3.3 ENUMERATING BEST-RESPONSE REGIONS AND COMPUTING THE OFFLINE OPTIMAL

Let $\mathcal{W} = \{W \in \mathcal{A}^{n \times K} \mid R(W) \neq \emptyset\}$ denote the set of mappings $W$ for which the corresponding best-response region $R(W)$ is non-empty. Although the total number of $W \in \mathcal{A}^{n \times K}$ is $A^{n \times K}$, the number of *non-empty* best-response regions is significantly smaller, especially when $L$ (number of actions of the leader) is treated as a constant. The exact characterization is given below. The proof (in Appendix A) uses a result in computational geometry regarding the number of nonempty regions obtained by dividing $\mathbb{R}^L$ using $\mathcal{O}(nKA^2)$ hyperplanes.

**Lemma 3.2.** *The number of non-empty best-response regions, $|\mathcal{W}|$, is $\mathcal{O}(n^L K^L A^{2L})$.*

For any algorithm to leverage these best response regions, it is imperative that these regions can be enumerated efficiently. The following lemma shows this is always possible. Intuitively, we construct a graph where the nodes represent non-empty best-response regions and an edge exists between $W, W' \in \mathcal{W}$ if and only if $W$ and $W'$ differ in exactly one entry. Traversing an edge, therefore, corresponds to moving to an adjacent best-response region by crossing a single hyperplane boundary. We show that this graph is always connected and can thus be efficiently traversed using breadth-first search. The exact algorithm and proof of Lemma 3.3 are in Appendix A.

**Lemma 3.3.** *The set of non-empty best-response regions $\{R(W) : W \in \mathcal{W}\}$ can be enumerated in $\text{poly}(n^L, K^L, A^L, L)$ time.*

We now show that the optimal strategy within each region can be efficiently computed. Recall from Definition 2.2 that, when given the followers' type distribution $\mathcal{D}$, computing the leader's optimal strategy requires solving $\max_{x \in \Delta(\mathcal{L})} \mathbb{E}_{\boldsymbol{\theta} \sim \mathcal{D}}[u(x, \mathbf{br}(\boldsymbol{\theta}, x))]$. Since the leader's utility is linear within a region $R(W)$, the optimal solution within $R(W)$ can be computed by the following linear program:

$$\max_{x \in R(W)} \sum\nolimits_{\boldsymbol{\theta} \in \Theta^n} \mathcal{D}(\boldsymbol{\theta}) u(x, \mathbf{br}(\boldsymbol{\theta})) = \max_{x \in R(W)} \sum\nolimits_{\boldsymbol{\theta} \in \Theta^n} \mathcal{D}(\boldsymbol{\theta}) u(x, W(\boldsymbol{\theta})) \tag{1}$$

where $x \in R(W)$ is given by the following set of linear constraints:

$$\begin{cases} \sum_{\ell \in \mathcal{L}} x(\ell) \big[ v_i(\ell, w_i(\theta_i), \theta_i) - v_i(\ell, a_i', \theta_i) \big] \geq 0, & \forall i \in [n], \ \forall \theta_i \in \Theta, \ \forall a_i' \in \mathcal{A}, \\ x(\ell) \geq 0, \ \forall \ell \in \mathcal{L}, \ \text{and} \ \sum_{\ell \in \mathcal{L}} x(\ell) = 1. \end{cases} \tag{2}$$

While there are $\mathcal{O}(nKA)$ constraints, each involving a sum over $L$ elements, the objective involves summing over $K^n$ possible type profiles. While this is exponential in $n$, any input to the complete information instance must provide the joint type distribution $\mathcal{D} \in [0, 1]^{K^n}$ as input. Thus, the time to compute the optimal solution within each region is polynomial in the input size.

The above results imply that, given distribution $\mathcal{D}$, the optimal leader strategy in BSGs can be computed efficiently when the number $L$ of leader's actions is small. This is because the optimal strategy within each best-response region $R(W)$ can be computed efficiently by the linear program (1), the overall optimal strategy is the maximum over all non-empty best-response regions, and there are at most $\mathcal{O}(n^L K^L A^{2L})$ such regions by Lemma 3.2. We thus showed above that BSGs are polynomial-time solvable for a constant $L$. In comparison, Theorem 7 of Conitzer & Sandholm (2006) proves that the optimal strategy is NP-hard to compute with an asymptotically increasing $L$.

## 4 TYPE FEEDBACK

### 4.1 LEARNING ALGORITHMS AND UPPER BOUNDS

**General Type Distributions:** We now address the core problem of learning the optimal leader strategy from online feedback. This section considers the type-feedback setting, where the leader observes each follower's realized type $\boldsymbol{\theta}^t = (\theta_1^t, \ldots, \theta_n^t)$ at the end of round $t$. We start with general distributions – that is, the followers' types can be arbitrarily correlated. Observing types after each round allows us to directly estimate the unknown distribution $\mathcal{D}$ and compute an optimal strategy accordingly. This is formalized in Algorithm 1:

---

**ALGORITHM 1:** Type-Feedback Algorithm – General Type Distributions

At round $t = 1$, pick an arbitrary strategy $x^1 \in \Delta(\mathcal{L})$.
**for** *round $t \geq 2$* **do**
$\quad$ Choose $x^t \in \arg\max_{x \in \Delta(\mathcal{L})} \sum_{s=1}^{t-1} u(x, \mathbf{br}(\boldsymbol{\theta}^s, x))$ – the empirically optimal strategy.
$\quad$ Observe the followers' types $\boldsymbol{\theta}^t \sim \mathcal{D}$.

---

At first glance, one might think that this algorithm might suffer a large regret because the distribution $\mathcal{D}$, which has support size $|\Theta^n| = K^n$, is difficult to estimate. Indeed, the estimation error for such a distribution using $t$ samples is at least $\Omega\big(\sqrt{\frac{K^n}{t}}\big)$ even if $\mathcal{D}$ is a product distribution (namely, the types are independent) (Lin, 2022). This suggests that the empirically optimal strategy $x^t$ might be worse than the true optimal strategy $x^*$ by at least $\Omega\big(\sqrt{\frac{K^n}{t}}\big)$, which would cause an $\Omega(\sqrt{K^n T})$ regret in $T$ rounds in total. As we will show in Theorem 4.1, one analysis of Algorithm 1 achieves exactly this as a regret upper bound. The proof (in Appendix B.2) upper bounds the single-round regret by the total variation (TV) distance between the empirical distribution $\hat{\mathcal{D}}^t$ and the true distribution $\mathcal{D}$.

While this suggests that $\mathcal{O}(\sqrt{K^n T})$ regret might be tight, this is interestingly not true when $n$ is large! That is, the intuitive lower bound that arises from the estimation error for distribution $\mathcal{D}$ is not correct. Although the empirical type distribution can differ significantly from the true type distribution, the empirical *utility* of any strategy $x \in \Delta(\mathcal{L})$ is actually concentrated around the true expected utility of $x$ with high probability. We formalize this below:

**Lemma 4.1.** *Given $t$ samples $\boldsymbol{\theta}^1, \ldots, \boldsymbol{\theta}^t$ from distribution $\mathcal{D}$, let $\hat{U}^t(x) = \frac{1}{t} \sum_{s=1}^t u(x, \mathbf{br}(\boldsymbol{\theta}^s, x))$ be the empirical expected utility of a strategy $x \in \Delta(\mathcal{L})$ computed on the $t$ samples. Recall that $U_{\mathcal{D}}(x) = \mathbb{E}_{\boldsymbol{\theta} \sim \mathcal{D}}[u(x, \mathbf{br}(\boldsymbol{\theta}, x))]$ denotes the true expected utility of $x$. With probability at least $1 - \delta$, we have: for all $x \in \Delta(\mathcal{L})$, $\big|U_{\mathcal{D}}(x) - \hat{U}^t(x)\big| \leq \mathcal{O}\big(\sqrt{\frac{L \log t}{t}} + \sqrt{\frac{L \log(nKA) + \log(1/\delta)}{t}}\big).$*

*Proof sketch.* By Lemma 3.2, the leader's strategy space $\Delta(\mathcal{L})$ can be divided into $|\mathcal{W}| = \mathcal{O}(n^L K^L A^{2L})$ best-response regions, and the leader's utility function $U_{\mathcal{D}}(x)$ is linear inside each region (Lemma 3.1). Because the *pseudo-dimension* of linear functions in an $L$-dimensional space are at most $L$, we have with probability at least $1 - \delta'$, the empirical utility $\hat{U}^t(x)$ on $t$ samples approximates the true expected utility $U_{\mathcal{D}}(x)$ with accuracy $\mathcal{O}\left(\sqrt{\frac{L \log t}{t}} + \sqrt{\frac{\log(1/\delta')}{t}}\right)$ for every strategy $x$ inside a best-response region. Taking a union bound over all $\mathcal{O}(n^L K^L A^{2L})$ best-response regions, i.e., letting $\delta' = \delta / \mathcal{O}(n^L K^L A^{2L})$, proves the lemma. See details in Appendix B.1. □

Note that the above concentration result holds for all strategies $x \in \Delta(\mathcal{L})$ simultaneously, instead of for a single fixed strategy (which easily follows from Hoeffding's inequality). This result means that the simple Algorithm 1 can achieve a regret that is of the order $\sqrt{T}$, logarithmic in $n$, with an additional $\sqrt{L}$ factor. This is better for large $n$ and small $L$. This new regret bound, along with the earlier one $\mathcal{O}(\sqrt{K^n T})$, is formalized in Theorem 4.1 below, with the proof given in Appendix B.2.

**Theorem 4.1.** *The type-feedback Algorithm 1 for general type distributions achieves expected regret* $\mathcal{O}\left(\min\left\{\sqrt{LT \cdot \log(nKAT)}, \sqrt{K^n T}\right\}\right)$ *and can be implemented in* $\text{poly}((nKA)^L LT)$ *time.*

Theorem 4.1 also comments on the runtime of Algorithm 1, which hinges on the computability of $x^t \in \arg\max_{x \in \Delta(\mathcal{L})} \sum_{s=1}^{t-1} u(x, \mathbf{br}(\boldsymbol{\theta}^s, x))$. Using the techniques developed in Section 3, this maximization can be solved by taking the maximum over the optimal strategies from each non-empty best-response region $W \in \mathcal{W}$, computed using the empirical type distribution. Using Lemmas 3.2 and 3.3 and the fact that the optimal strategy within a non-empty $R(W)$ can be solved by the following linear program, we obtain a runtime that is polynomial when $L$ is constant:[4]

$$\max_{W \in \mathcal{W}} \left\{ \max_{x \in R(W)} \sum_{s=1}^{t} u(x, W(\boldsymbol{\theta}^s)) \text{ subject to the constraints in (2)} \right\}. \tag{3}$$

**Independent Type Distributions:** Algorithm 1 and the corresponding regret bound in Theorem 4.1 hold without any assumptions on the joint type distribution $\mathcal{D}$. In many settings, however, the followers' types may be independent of one another. Intuitively, one expects learning to be easier in such settings since it suffices to learn the marginals as opposed to the richer joint distribution. This is indeed correct: in Algorithm 2, we build the empirical distribution $\hat{\mathcal{D}}_i^t$ for each marginal from samples $\theta_i^1, \ldots, \theta_i^t$ for follower $i$ and then take the product $\hat{\boldsymbol{\mathcal{D}}}^t = \prod_{i=1}^{n} \hat{\mathcal{D}}_i^t$ to estimate $\boldsymbol{\mathcal{D}} = \prod_{i=1}^{n} \mathcal{D}_i$.

---

**ALGORITHM 2:** Type-Feedback Algorithm - Independent Type Distributions

At $t = 1$, pick an arbitrary strategy $x^1 \in \Delta(\mathcal{L})$.
**for** *round $t > 1$* **do**
  Choose $x^t \in \arg\max_{x \in \Delta(\mathcal{L})} \mathbb{E}_{\boldsymbol{\theta} \sim \hat{\boldsymbol{\mathcal{D}}}^{t-1}}[u(x, \mathbf{br}(\boldsymbol{\theta}, x))]$
  Observe realized follower type $(\theta_1^t, \ldots, \theta_n^t)$
  **for** $i \in [n]$, $k \in \Theta$ **do**
    $\hat{\mathcal{D}}_i^t(k) = \frac{1}{t} \sum_{s=1}^{t} \mathbb{1}[\theta_i^s = k]$
  $\hat{\boldsymbol{\mathcal{D}}}^t(\boldsymbol{\theta}) = \prod_{i=1}^{n} \hat{\mathcal{D}}_i^t(\theta_i), \forall \boldsymbol{\theta} \in \Theta^n$

---

This algorithm achieves a much improved regret, $\mathcal{O}(\sqrt{nKT})$, formalized in Theorem 4.2 and empirically verified in Appendix D. The proof (in Appendix B.3) is similar to the $\mathcal{O}(\sqrt{K^n T})$ regret analysis of Theorem 4.1, which upper bounds the single-round regret by the TV distance between $\hat{\boldsymbol{\mathcal{D}}}^t$ and $\mathcal{D}$. But for independent distributions, we can relate the TV distance with the sum of Hellinger distances between the marginals $\hat{\mathcal{D}}_i^t$ and $\mathcal{D}_i$, which is bounded by $\mathcal{O}(\sqrt{\frac{nK}{t}})$ instead of $\mathcal{O}(\sqrt{\frac{K^n}{t}})$, so the total regret is bounded by $\mathcal{O}(\sqrt{nKT})$. The computational complexity, though, increases to $\text{poly}((nKA)^L LT K^n)$ as the empirical product distribution $\hat{\boldsymbol{\mathcal{D}}}^t = \prod_i^n \hat{\mathcal{D}}_i^t$ has support size $K^n$.

---

[4]Also note that Algorithm 1 does not need as input the entire utility function of the leader $u(\cdot, \cdot)$, which has an exponential size $L \cdot A^n$. The algorithm only needs the utility function for the sampled types.

**Theorem 4.2.** *The type-feedback Algorithm 2 for independent type distributions achieves expected regret $\mathcal{O}\big(\sqrt{nKT}\big)$ and can be implemented in $\text{poly}((nKA)^L LTK^n)$ time.*

**Corollary 4.1.** *Taking the minimum of Theorems 4.1 and 4.2, we obtain a type-feedback algorithm with expected regret $\mathcal{O}\big(\min\big\{\sqrt{LT \cdot \log(nKAT)}, \sqrt{nKT}\big\}\big)$ for independent type distributions.*

## 4.2 LOWER BOUND

We then provide a lower bound result: no algorithm for online Bayesian Stackelberg game has a better regret than $\Omega(\sqrt{\min\{L, nK\}T})$. When the number of followers $n$ is large, this lower bound matches the previous upper bounds $\widetilde{\mathcal{O}}(\sqrt{LT})$. To our knowledge, this work is the first to provide a lower bound for the multi-follower problem and give an almost tight characterization of the factor before the classical $\sqrt{T}$ term. Interestingly, this $\widetilde{\mathcal{O}}(\sqrt{L})$ factor does not grow with $n$ up to log factor.

**Theorem 4.3.** *The expected regret of any type-feedback algorithm is at least $\Omega(\sqrt{\min\{L, nK\}T})$. This holds even if the followers' types are independent and the leader's utility does not depend $\ell$.*

The proof (given in Appendix B.5) involves two non-trivial reductions. First, we reduce the *distribution learning* problem to a *single-follower* Bayesian Stackelberg game, obtaining an $\Omega(\sqrt{\min\{L, K\}T})$ lower bound. Then, we reduce the single-follower game with $nK$ types to a game with $n$ followers each with $K$ types. One might wish to reduce a single-follower game with $K^n$ types to an $n$-follower game to prove a lower bound of $\Omega(\sqrt{\min\{L, K^n\}T})$ for general type distributions, but that cannot be done easily due to no externality between the followers.

## 5 ACTION FEEDBACK

We now discuss the setting where the leader observes the followers' actions after each round. This setting is more practical yet challenging than the type-feedback setting. We present two learning algorithms. The first algorithm achieves $\mathcal{O}(K^n\sqrt{T}\log T)$ regret, using a previous technique from Bernasconi et al. (2023). The second algorithm involves a novel combination of the Upper Confidence Bound principle and the concentration analysis of best-response regions from Lemma 4.1, achieving $\mathcal{O}(\sqrt{n^L K^L A^{2L} LT \log T})$ regret. The latter is better when the number of followers $n$ is large and the number of leader actions $L$ is small. We empirically simulate both approaches in Appendix D.

**Linear-bandit based approach with $\mathcal{O}(K^n\sqrt{T}\log T)$ regret:** Bernasconi et al. (2023) developed a technique to reduce the online learning problem of solving a linear program with unknown objective to a linear bandit instance. A spiritually similar approach can be applied here. While the optimization problem for our Bayesian Stackelberg games (Definition 2.2) is not a linear program, we show that under a different formulation, this can actually be solved by a single linear program (we explain the details in the proof of Theorem 5.1). We can thus leverage the techniques of Bernasconi et al. (2023) to reduce this to a linear bandit problem. Since Bernasconi et al. (2023) considers an adversarial online learning environment (ours is stochastic), directly applying their technique will lead to a sub-optimal $\widetilde{\mathcal{O}}(K^{\frac{3n}{2}}\sqrt{T})$ regret bound. Instead, we apply the OFUL algorithm for stochastic linear bandit (Abbasi-yadkori et al., 2011) to obtain a better regret bound of $\widetilde{\mathcal{O}}(K^n\sqrt{T})$. Instead, we apply the OFUL algorithm for stochastic linear bandit (Abbasi-yadkori et al., 2011) to obtain a better regret bound of $\widetilde{\mathcal{O}}(K^n\sqrt{T})$. See details in Appendix C.1.

**Theorem 5.1.** *There exists an action-feedback algorithm for online Bayesian Stackelberg game with $\mathcal{O}(K^n\sqrt{T}\log T)$ regret.*

**Algorithm 3 with $O(\sqrt{n^L K^L A^{2L} LT \log T})$ regret.** We design a better algorithm for large $n$ and small $L$, not using Bernasconi et al. (2023)'s technique but using the "concentration over best-response regions" idea we developed in the previous sections. Recall from Section 3 that the leader's strategy space can be partitioned into best-response regions: $\Delta(\mathcal{L}) = \bigcup_{W \in \mathcal{W}} R(W)$. When the leader plays strategy $x$ in a region $R(W)$, the followers' best-response function satisfies $\mathbf{br}(\boldsymbol{\theta}, x) = W(\boldsymbol{\theta})$, so

the leader's expected utility is

$$U(x, R(W)) = \sum_{\boldsymbol{\theta} \in \Theta^n} \boldsymbol{\mathcal{D}}(\boldsymbol{\theta}) u(x, W(\boldsymbol{\theta})) = \sum_{\boldsymbol{a} \in \mathcal{A}^n} u(x, \boldsymbol{a}) \sum_{\boldsymbol{\theta} | W(\boldsymbol{\theta}) = \boldsymbol{a}} \boldsymbol{\mathcal{D}}(\boldsymbol{\theta}) = \sum_{\boldsymbol{a} \in \mathcal{A}^n} u(x, \boldsymbol{a}) \mathcal{P}(\boldsymbol{a} \mid R(W))$$

where $\mathcal{P}(\boldsymbol{a} \mid R(W)) = \sum_{\boldsymbol{\theta} \in \Theta^n : W(\boldsymbol{\theta}) = \boldsymbol{a}} \boldsymbol{\mathcal{D}}(\boldsymbol{\theta})$ denotes the probability that the followers jointly take action $\boldsymbol{a} \in \mathcal{A}^n$ when the leader plays $x \in R(W)$. Since the distribution $\mathcal{P}(\cdot \mid R(W)) \in \Delta(\mathcal{A}^n)$ does not depend on $x$ as long as $x \in R(W)$, playing $N$ strategies $x^1, ..., x^N$ within $R(W)$ yields $N$ observations $\boldsymbol{a}^1, ..., \boldsymbol{a}^N \sim \mathcal{P}(\cdot \mid R(W))$. Using these samples, we can estimate the utility of any other strategy $x \in R(W)$ within the same region. We define the empirical utility estimate on $N$ samples of joint actions as $\hat{U}_N(x, R(W)) = \frac{1}{N} \sum_{s=1}^{N} u(x, \boldsymbol{a}^s)$.

**Lemma 5.1.** *Suppose $T \geq |\mathcal{W}|$. With probability at least $1 - \frac{1}{T^2}$, we have:* $\forall W \in \mathcal{W}, \forall N \in \{1, \ldots, T\}, \forall x \in R(W), |U(x, R(W)) - \hat{U}_N(x, R(W))| \leq \sqrt{\frac{4(L+1)\log(3T)}{N}}.$

The proof of this lemma is similar to the proof of Lemma 4.1 and given in Appendix C.2.

For each region $W \in \mathcal{W}$, let $N^t(W) = \sum_{s=1}^{t-1} \mathbb{1}\left[x^s \in R(W)\right]$ be the number of times when strategies in region $R(W)$ were played in the first $t - 1$ rounds. Given the result in Lemma 5.1, we define an Upper Confidence Bound (UCB) on the expected utility of the optimal strategy in region $R(W)$:

$$\text{UCB}^t(W) = \max_{x \in R(W)} \left\{ \hat{U}_{N^t(W)}(x, R(W)) \right\} + \sqrt{\frac{4(L+1)\log(3T)}{N^t(W)}}.$$

We design the following algorithm: at each round $t$, select the region $W \in \mathcal{W}$ with the highest $\text{UCB}^t(W)$, play the empirically optimal strategy in that region, and increment $N^t(W)$ by 1. Full description of the algorithm is given in Algorithm 3.

---

**ALGORITHM 3:** Upper Confidence Bound (UCB) for Best-Response Regions

---
Let $\mathcal{W} = \{W \mid R(W) \neq \emptyset\}$.
**for** $W \in \mathcal{W}$ **do**
    Choose any strategy $x \in R(W)$ and observe a joint action.
**for** *round* $t > |\mathcal{W}|$ **do**
    **for** *each* $W \in \mathcal{W}$ **do**
        Let $N^t(W) = \sum_{s=1}^{t-1} \mathbb{1}\left[W^s = W\right]$ be the number of times region $R(W)$ was chosen.
        Let $\hat{\mathcal{P}}^t(\cdot \mid R(W))$ be the empirical distribution of joint actions in the rounds where region
        $R(W)$ was chosen: $\hat{\mathcal{P}}^t(\boldsymbol{a} \mid R(W)) = \frac{1}{N^t(W)} \sum_{s=1}^{t-1} \mathbb{1}\left[W^s = W\right] \cdot \mathbb{1}\left[\boldsymbol{a}^s = \boldsymbol{a}\right].$
        Compute the empirically optimal strategy in region $R(W)$:

$$\hat{x}^*_{R(W)} = \arg\max_{x \in R(W)} \mathbb{E}_{\boldsymbol{a} \sim \hat{\mathcal{P}}^t(\cdot \mid R(W))}\left[u(x, \boldsymbol{a})\right],$$

        which has empirical utility $\hat{u}^*_{R(W)} = \mathbb{E}_{\boldsymbol{a} \sim \hat{\mathcal{P}}^t(\cdot \mid R(W))}\left[u(\hat{x}^*_{R(W)}, \boldsymbol{a})\right].$
        Let $\text{UCB}^t(W) = \hat{u}^*_{R(W)} + \sqrt{\frac{4(L+1)\log(3T)}{N^t(W)}}.$
    Let $W^t \in \arg\max_{W \in \mathcal{W}} \text{UCB}^t(W).$
    Play strategy $x^t = \hat{x}^*_{R(W^t)}$ and observe joint action $\boldsymbol{a}^t = (a_1^t, \ldots, a_n^t).$

---

**Theorem 5.2.** *Algorithm 3 has expected regret* $\mathcal{O}\left(\sqrt{n^L K^L A^{2L} L \cdot T \log T}\right).$

While the full proof is in Appendix C.3, we sketch the intuition. In the classical multi-armed bandit problem, the UCB algorithm has expected regret $\mathcal{O}(\sqrt{mT \log T})$ where $m$ is the number of arms. In our setting, each best-response region corresponds to an arm, and the confidence bound for each region is $\mathcal{O}(\sqrt{\frac{L \log T}{N^t(W)}})$. The number of arms/regions is $m = |\mathcal{W}| = \mathcal{O}(n^L K^L A^{2L})$ by Lemma 3.2. So, the regret of Algorithm 3 is at most $\mathcal{O}(\sqrt{|\mathcal{W}| \cdot L \cdot T \log T}) = \mathcal{O}(\sqrt{n^L K^L A^{2L} \cdot L \cdot T \log T}).$

**Corollary 5.1.** *By taking the better algorithm in Theorems 5.1 and 5.2, we obtain an action-feedback algorithm with* $\widetilde{\mathcal{O}}\left(\min\left\{K^n, \sqrt{n^L K^L A^{2L} L}\right\}\sqrt{T}\right)$ *regret.*

**Dependencies on various parameters:** Since action-feedback is more limited than type-feedback, the lower bound in Theorem 4.3 immediately carries over and shows that the $\widetilde{\mathcal{O}}(\sqrt{T})$ regret bounds here are tight in $T$. There are several subtleties in achieving tighter bounds on the remaining parameters. Conitzer & Sandholm (2006) show that, even with known distributions, BSG are NP-Hard to solve with respect to $L$; so an exponential computational dependence on $L$ is unavoidable, even if the regret could be made independent of $L$ as shown in our $\mathcal{O}(K^n\sqrt{T}\log T)$ result. Whether an online learning algorithm with $\text{poly}(n, K, L)\sqrt{T}$ regret exists is an open question, but such an algorithm will suffer an exponential runtime in $L$ unless P = NP.

## 6 Discussion

This work designed online learning algorithms for Bayesian Stackelberg games with multiple followers with unknown type distributions. Although the joint type space of $n$ followers has an exponentially large size $K^n$, we achieved significantly smaller regrets: $\widetilde{\mathcal{O}}(\sqrt{\min\{nK, L\}T})$ when the followers' types are independent and observable, $\widetilde{\mathcal{O}}(\sqrt{\min\{K^n, L\}T})$ when followers' types are correlated and observable, and $\widetilde{\mathcal{O}}(\min\{K^n, \sqrt{n^L K^L A^{2L} L}\}\sqrt{T})$ when only the followers' actions are observed. These results exploit various geometric properties of the leader's strategy space. The type-feedback bounds are tight in all parameters and the action-feedback bounds are tight in $T$. The exponential dependency on $L$ is unavoidable computationally (Conitzer & Sandholm, 2006). Further closing the gaps between upper and lower regret bounds is an open question and will likely involve tradeoff between different parameters and tradeoff between computation and regret.

### 6.1 Connections to the Adversarial Setting

Our work considers stochastic follower types. Alternatively, one can consider a setting where followers' types are adversarially generated. We conjecture that the main technique behind our results, "concentration over best response regions", can be generalized to adversarial settings. Importantly, the leader's optimization problem within each best-response region is linear. With stochastic follower types, the linear optimization problem within each best-response region can be solved by empirical optimization – this algorithm needs to be changed in the adversarial setting. In more detail:

- For the type-feedback case, we can run an adversarial full-feedback multi-armed bandit algorithm (e.g., Multiplicative Weights Update) to pick a best-response region at each round. Within each best-response region, the problem becomes an adversarial online linear optimization problem with full feeback, to which we can apply algorithms such as FTRL to choose leader strategies.
- For action-feedback, we can run an adversarial partial-feedback MAB algorithm (e.g., EXP3) to pick a best-response region at each round. Within each region, use an adversarial partial-feedback online linear optimization algorithm to choose leader strategies.

Intuitively, best-response regions allow us (1) to tame the continuous leader action space and (2) to reduce the problem to classical adversarial problems. Formally verifying this approach is left open.

### 6.2 Inter-Follower Externality

Our work assumed no externality between the followers. While the learning algorithms for multiple followers without externality may not differ significantly from that of a single follower, one might expect the regret bound to grow at the exponential rate of $O(\sqrt{K^n T})$. However, we show that the regret bounds can be significantly lower when $L$ is smaller than $n$ (Theorems 4.1, 5.2).

The presence of inter-follower externality leads to a leader-strategy-induced simultaneous game among the followers. Putting aside the computational difficulties of inter-follower Nash Equilibrium (or assuming oracle access to it), some of our results in the type-feedback setting generalize. In particular, our $O(\sqrt{K^n T})$ result for general type distributions and $O(\sqrt{nKT})$ result for independent type distributions also apply to the setting with inter-follower externality. Our other results, which are built on best-response region characterization and depend on the independence of followers' utility functions, do not generalize to the case with externality. The introduction of inter-follower externality significantly complicates the design and analysis of learning algorithms, because the action feedback now comes from equilibrium responses, instead of individual best responses to leader strategy.

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

# A  APPENDIX FOR SECTION 3

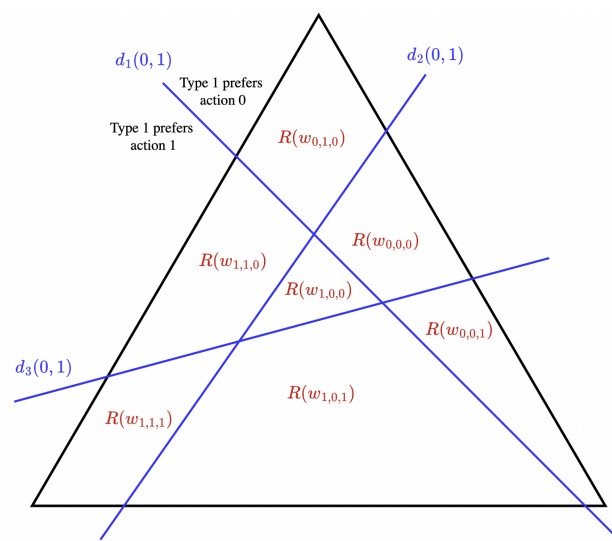

Figure 1: A single-follower best-response region with $K = 3$ types and two follower actions and three leader actions – $A = 2, L = 3$. The triangle represents the probability simplex $\Delta(\mathcal{L})$. The three hyperplanes defined by $d_1(0,1)$, $d_2(1,0)$ and $d_3(1,0)$ partition the simplex into best-response regions. For example, in region $R(w_{0,1,1})$, the follower best-responds with action $0$ for type $1$, and action $1$ for types $2$ and $3$.

## A.1  PROOF OF LEMMA 3.2

*Proof.* We have $n$ followers each with $K$ type and $A$ actions. Each follower has $K\binom{A}{2} \leq KA^2$ advantage vectors, where each advantage vector $d_{\theta,a,a'}$ corresponds to a hyperplane in $\mathbb{R}^L$ that separates the leader's mixed strategy space $\Delta(\mathcal{L}) \subseteq \mathbb{R}^L$ into two halfspaces. In total, $n$ followers have $nKA^2$ hyperplanes. Those hyperplanes divide $\mathbb{R}^L$ into at most $\mathcal{O}((nKA^2)^L) = \mathcal{O}(n^L K^L A^{2L})$ regions (see, e.g., Halperin & Sharir (2017)). Each non-empty best response region is one of such regions, so the total number is $\mathcal{O}(n^L K^L A^{2L})$. □

## A.2  PROOF OF LEMMA 3.3

---

**ALGORITHM 4:** Best-Response Region Enumeration

---

Let isFeasible$(x, i, \theta, a) = \mathbb{1}\left[\forall a' \in \mathcal{A}, \sum_{\ell \in \mathcal{L}} x(\ell)\big(v_i(\ell, a, \theta) - v_i(\ell, a', \theta)\big) \geq 0\right]$
Let findFeasible$(W) = \big\{x \in \Delta(\mathcal{L}) \,|\, \forall i \in [n], \theta \in \Theta,\, \text{isFeasible}(x, i, \theta, W[i, \theta]) = 1\big\}$
Choose a random strategy $x_{init} \in \Delta(\mathcal{L})$
**for** $i \in [n]$, $\theta \in \Theta$ **do**
    **for** $a \in \mathcal{A}$ **do**
        **if** *isFeasible*$(x_{init}, i, \theta, a)$ **then**
            $W_{init}[i, \theta] = a$

Let $queue = [W_{init}]$,    mark $W_{init}$ as *visited*
**while** $queue \neq \emptyset$ **do**
    $W = queue.pop()$
    **for** $i \in [n]$, $\theta \in \Theta$ **do**
        **for** $a \in \mathcal{A}$ *and* $a \neq W[i, \theta]$ **do**
            Let $W' = W$
            Let $W'[i, \theta] = a$
            **if** *findFeasible*$(W) \neq \emptyset$ *and* $W'$ *is not visited* **then**
                $queue.append(W')$,    mark $W'$ as *visited*

---

*Proof.* We construct a graph $G = (V, E)$ where $V$ consists of the elements $W$, each representing a best-response region. An edge exists between two vertices $W, W' \in \mathcal{W}$ if and only if $W$ and $W'$

differ in exactly one entry. Traversing an edge corresponds to moving between adjacent best-response regions by crossing a hyperplane boundary.

We claim that this graph is connected. Since each vertex $W$ corresponds to a best-response region defined by the inequalities in (2), and the leader's strategy space is the $L$-dimensional probability simplex, the union of non-empty best response regions forms a partition of the strategy space. Because these regions are convex polytopes sharing boundaries, the adjacent structure defined by differing in one entry corresponds to crossing a shared facet. Starting from any non-empty region, we can traverse to any other by crossing shared facets through adjacent regions, so the graph is connected.

Thus, to enumerate all non-empty best-response regions, we can perform a graph search (e.g., breadth-first search or depth-first search) starting from any initial vertex $W$ to traverse all vertices in $\mathcal{O}(|W|)$ steps, which is $\mathcal{O}(n^L K^L A^{2L})$ by Lemma 3.2. Specifically, at each vertex $W$, we examine all its adjacent $nKA$ vertices. For each adjacent vertex $W'$, we determine whether $R(W')$ is a non-empty region by solving a feasibility linear program defined by the constraints in (2), which runs in $\mathrm{poly}(n, K, A, L)$ time. Then, the total running time is $\mathrm{poly}(n^L, K^L, A^L, L)$. We present the algorithm formally in Algorithm 4. $\qquad\square$

## B  Appendix for Section 4

The following definitions will be used in the proofs for this section:

**Definition B.1** (Total variation distance). *For two discrete distributions $\mathcal{D}$ and $\hat{\mathcal{D}}$ over support $\Theta$, the total variation is half the $L_1$ distance between the two distributions:*

$$\delta(\mathcal{D}, \hat{\mathcal{D}}) = \frac{1}{2}||\mathcal{D} - \hat{\mathcal{D}}||_1 = \frac{1}{2}\sum_{\theta \in \Theta}|\mathcal{D}(\theta) - \hat{\mathcal{D}}(\theta)|.$$

**Definition B.2** (Hellinger distance). *For two discrete distributions $\mathcal{D}$ and $\hat{\mathcal{D}}$ over support $\Theta$, the Hellinger distance is defined as*

$$H(\mathcal{D}, \hat{\mathcal{D}}) = \frac{1}{\sqrt{2}}\|\sqrt{\mathcal{D}} - \sqrt{\hat{\mathcal{D}}}\|_2 = \frac{1}{\sqrt{2}}\sqrt{\sum_{\theta \in \Theta}\left(\mathcal{D}(\theta) - \hat{\mathcal{D}}(\theta)\right)^2}.$$

### B.1  Proof of Lemma 4.1

We rely on several key insights about the *pseudo-dimension* of a family of functions, defined below:

**Definition B.3** (Definition 10.2 in Mohri et al. (2012)). *Let $\mathcal{G}$ be a family of functions from input space $\mathcal{Z}$ to real numbers $\mathbb{R}$.*

- *A set of inputs $\{z_1, \ldots, z_m\} \subseteq \mathcal{Z}$ is* shattered *by $\mathcal{G}$ if there exists thresholds $t_1, \ldots, t_m \in \mathbb{R}$ such that for any sign vector $\boldsymbol{\sigma} = (\sigma_1, \ldots, \sigma_m) \in \{-1, +1\}^m$, there exists a function $g \in \mathcal{G}$ satisfying $\mathrm{sign}(g(x_i) - t_i) = \sigma_i$ for all $i = 1, \ldots m$.*
- *The size of the largest set of inputs that can be shattered by $\mathcal{G}$ is called the* pseudo-dimension *of $\mathcal{G}$, denoted by $\mathrm{Pdim}(\mathcal{G})$.*

Given a family of functions with a finite pseudo-dimension, and samples $z^1, \ldots, z^N$ drawn from a distribution on the input space $\mathcal{Z}$, the empirical mean of any function in the family will, with high probability, be close to the true mean. Formally:

**Theorem B.1** (e.g., Theorem 10.6 in Mohri et al. (2012)). *Let $\mathcal{G}$ be a family of functions from $\mathcal{Z}$ to $[0, 1]$ with pseudo-dimension $\mathrm{Pdim}(\mathcal{G}) = d$. For any distribution $F$ over $\mathcal{Z}$, with probability at least $1 - \delta$ over the random draw of $N$ samples $z^1, \ldots, z^N$ from $F$, the following holds for all $g \in \mathcal{G}$,*

$$\left|\mathbb{E}_{z \sim F}[g(z)] - \frac{1}{N}\sum_{i=1}^{N}g(z^i)\right| \leq \sqrt{\frac{2d \log 3N}{N}} + \sqrt{\frac{\log\frac{1}{\delta}}{2N}}.$$

Consider the family of linear functions over $\mathbb{R}^L$: $\mathcal{G} = \{g_x : z \to \langle x, z \rangle \mid x \in \mathbb{R}^L\}$. It is known that the pseudo-dimension of this family is $L$:

**Lemma B.1** (e.g., Theorem 10.4 in Mohri et al. (2012)). *The family of linear functions $\{g_x : z \to \langle x, z \rangle \mid x \in \mathbb{R}^L\}$ in $\mathbb{R}^L$ has pseudo-dimension L.*

*Proof of Lemma 4.1.* We now have the tools to prove Lemma 4.1. For any non-empty best-response region defined by $W \in \mathcal{W}$, let $\boldsymbol{\theta}^1, ..., \boldsymbol{\theta}^t \sim \mathcal{D}$ be $t$ i.i.d samples. For each sample $\boldsymbol{\theta}^i$, we can directly compute

$$z_{W,\boldsymbol{\theta}^i} = \Big( u(1, W(\boldsymbol{\theta}^i)), ..., u(L, W(\boldsymbol{\theta}^i)) \Big).$$

Fix any leader strategy $x \in R(W)$. By Lemma 3.1, the leader's expected utility by using strategy $x$ is $U_{\mathcal{D}}(x) = \mathbb{E}_{\boldsymbol{\theta} \sim \mathcal{D}}[\langle x, z_{W,\boldsymbol{\theta}} \rangle]$, which is the expectation of the linear function $g_x(z_{W,\boldsymbol{\theta}}) = \langle x, z_{W,\boldsymbol{\theta}} \rangle$. Therefore, by Theorem B.1 and Lemma B.1, we have

$$\Pr\left[ \forall x \in R(W), \; \Big| \mathbb{E}_{\boldsymbol{\theta} \sim \mathcal{D}}[g_x(z_{W,\boldsymbol{\theta}})] - \frac{1}{t} \sum_{i=1}^{t} g_x(z_{W,\boldsymbol{\theta}^i}) \Big| \le \sqrt{\frac{2L \log 3t}{t}} + \sqrt{\frac{\log \frac{1}{\delta'}}{2t}} \right] \ge 1 - \delta'.$$

By definition, $U_{\mathcal{D}}(x) = \mathbb{E}_{\boldsymbol{\theta} \sim \mathcal{D}}[g_x(z_{W,\boldsymbol{\theta}})]$ and $\hat{U}^t(x) = \frac{1}{t} \sum_{i=1}^{t} g(z_{W,\boldsymbol{\theta}^t})$. So,

$$\Pr\left[ \forall x \in R(W), \; \Big| U_{\mathcal{D}}(x) - \hat{U}^t(x) \Big| \le \sqrt{\frac{2L \log 3t}{t}} + \sqrt{\frac{\log \frac{1}{\delta'}}{2t}} \right] \ge 1 - \delta'.$$

Let $\delta' = \frac{\delta}{|\mathcal{W}|}$. By the union bound, with probability at least $1 - \delta$, the following bound holds simultaneously for all $W \in \mathcal{W}$ and $x \in R(W)$:

$$\Big| U_{\mathcal{D}}(x) - \hat{U}^t(x) \Big| \le \sqrt{\frac{2L \log(3t)}{t}} + \sqrt{\frac{\log \frac{|\mathcal{W}|}{\delta}}{2t}} = \mathcal{O}\left( \sqrt{\frac{L \log t}{t}} + \sqrt{\frac{L \log(nKA) + \log(\frac{1}{\delta})}{t}} \right),$$

where we used the fact $|\mathcal{W}| = O(n^L K^L A^{2L})$ from Lemma 3.2. □

## B.2 PROOF OF THEOREM 4.1

**Analysis of $\mathcal{O}(\sqrt{K^n T})$ Regret:** Consider a Bayesian Stackelberg game with $n$ followers each with $K$ types, with joint type distribution $\mathcal{D}$. Let $U(x, \mathcal{D}) = \mathbb{E}_{\boldsymbol{\theta} \sim \mathcal{D}}[u(x, \mathbf{br}(\boldsymbol{\theta}, x))]$ be the expected utility of the leader playing mixed strategy $x$ when the type distribution is $\mathcal{D}$. Let $x^* = \arg\max_{x \in \Delta(\mathcal{L})} U(x, \mathcal{D})$ be the optimal strategy for $\mathcal{D}$. At each round $t$, Algorithm 1 chooses the optimal strategy $x^t = \arg\max_{x \in \Delta(\mathcal{L})} U(x, \hat{\mathcal{D}}^{t-1})$ for the empirical distribution $\hat{\mathcal{D}}^{t-1}$ over $t - 1$ samples. The total expected regret is equal to

$$\mathbb{E}[\text{Reg}(T)] = \sum_{t=1}^{T} \mathbb{E}\Big[ \underbrace{U(x^*, \mathcal{D}) - U(x^t, \mathcal{D})}_{\text{single-round regret } r(t)} \Big],$$

We upper bound the single-round regret $r(t)$ by the total variation distance (Definition B.1) between $\mathcal{D}$ and $\hat{\mathcal{D}}^{t-1}$:

**Claim B.1.** $r(t) = U(x^*, \mathcal{D}) - U(x^t, \mathcal{D}) \le 4\delta(\mathcal{D}, \hat{\mathcal{D}}^{t-1}).$

*Proof.*

$$r(t) = U(x^*, \mathcal{D}) - U(x^t, \mathcal{D})$$
$$= U(x^*, \mathcal{D}) - U(x^*, \hat{\mathcal{D}}^{t-1}) + U(x^*, \hat{\mathcal{D}}^{t-1}) - U(x^t, \hat{\mathcal{D}}^{t-1}) + U(x^t, \hat{\mathcal{D}}^{t-1}) - U(x^t, \mathcal{D})$$
$$\le U(x^*, \mathcal{D}) - U(x^*, \hat{\mathcal{D}}^{t-1}) + 0 + U(x^t, \hat{\mathcal{D}}^{t-1}) - U(x^t, \mathcal{D}) \tag{4}$$

where (4) follows from $U(x^*, \hat{\boldsymbol{\mathcal{D}}}^{t-1}) - U(x^t, \hat{\boldsymbol{\mathcal{D}}}^{t-1}) \leq 0$ because $x^t$ maximizes $U(x, \hat{\boldsymbol{\mathcal{D}}}^{t-1})$. We bound the first term in Equation (4) as follows:

$$\begin{aligned}
U(x^*, \boldsymbol{\mathcal{D}}) - U(x^*, \hat{\boldsymbol{\mathcal{D}}}^{t-1}) &= \sum_{\boldsymbol{\theta} \in \Theta} \boldsymbol{\mathcal{D}}(\boldsymbol{\theta}) \, u(x, \mathbf{br}(\boldsymbol{\theta}, x)) - \sum_{\boldsymbol{\theta} \in \Theta} \hat{\boldsymbol{\mathcal{D}}}^{t-1}(\boldsymbol{\theta}) \, u(x, \mathbf{br}(\boldsymbol{\theta}, x)) \\
&= \sum_{\boldsymbol{\theta} \in \Theta} \left( \boldsymbol{\mathcal{D}}(\boldsymbol{\theta}) - \hat{\boldsymbol{\mathcal{D}}}^{t-1}(\boldsymbol{\theta}) \right) u(x, \mathbf{br}(\boldsymbol{\theta}, x)) \\
&\leq \sum_{\boldsymbol{\theta} \in \Theta} \left| \boldsymbol{\mathcal{D}}(\boldsymbol{\theta}) - \hat{\boldsymbol{\mathcal{D}}}^{t-1}(\boldsymbol{\theta}) \right| u(x, \mathbf{br}(\boldsymbol{\theta}, x)) \\
&\leq \sum_{\boldsymbol{\theta} \in \Theta} \left| \boldsymbol{\mathcal{D}}(\boldsymbol{\theta}) - \hat{\boldsymbol{\mathcal{D}}}^{t-1}(\boldsymbol{\theta}) \right| \cdot 1 \; = \; 2\delta(\boldsymbol{\mathcal{D}}, \hat{\boldsymbol{\mathcal{D}}}^{t-1}).
\end{aligned}$$

By a symmetrical argument, the second term in Equation (4) is also bounded by $2\delta(\boldsymbol{\mathcal{D}}, \hat{\boldsymbol{\mathcal{D}}}^{t-1})$. $\quad\square$

Using Claim B.1 and taking expectation, we have

$$\mathbb{E}[r(t)] \; \leq \; 4\mathbb{E}[\delta(\boldsymbol{\mathcal{D}}, \hat{\boldsymbol{\mathcal{D}}}^{t-1})].$$

According to Canonne (2020), for distributions with support size $K^n$, $\mathbb{E}[\delta(\boldsymbol{\mathcal{D}}, \hat{\boldsymbol{\mathcal{D}}}^{t-1})] \leq \mathcal{O}(\sqrt{\frac{K^n}{t-1}})$. Thus, we have $\mathbb{E}[r(t)] \leq \mathcal{O}(\sqrt{\frac{K^n}{t-1}})$. Using the inequality $\sum_{t=1}^{T} \frac{1}{\sqrt{t}} \leq 2\sqrt{T}$, we obtain

$$\mathbb{E}[\mathrm{Reg}(T)] \; = \; \sum_{t=1}^{T} \mathbb{E}[r(t)] \; \leq \; \mathcal{O}\left( \sum_{t=1}^{T} \sqrt{\frac{K^n}{t}} \right) \; \leq \; \mathcal{O}(2\sqrt{K^n T}) \; = \; \mathcal{O}(\sqrt{K^n T}).$$

**Analysis of $\mathcal{O}(\sqrt{LT \log(nKAT)})$ Regret:** Consider round $t \geq 2$. By Lemma 4.1, we have that with probability at least $1 - \delta$:

$$\left| U_{\boldsymbol{\mathcal{D}}}(x) - \hat{U}^t(x) \right| \; \leq \; \mathcal{O}\left( \sqrt{\frac{L \log t}{t}} + \sqrt{\frac{L \log(nKA) + \log(\frac{1}{\delta})}{t}} \right), \quad \forall x \in \Delta(\mathcal{L}).$$

Suppose this event happens. Then, the regret of the algorithm at round $t$ is bounded as follows:

$$\begin{aligned}
r(t) &= \mathbb{E}_{\boldsymbol{\theta} \sim \boldsymbol{\mathcal{D}}} \left[ u(x^*, \mathbf{br}(\boldsymbol{\theta}, x^*)) - u(x^t, \mathbf{br}(\boldsymbol{\theta}, x^t)) \right] \\
&= U_{\boldsymbol{\mathcal{D}}}(x^*) - U_{\boldsymbol{\mathcal{D}}}(x^t) \\
&= U_{\boldsymbol{\mathcal{D}}}(x^*) - \hat{U}^{t-1}(x^*) + \hat{U}^{t-1}(x^*) - \hat{U}^{t-1}(x^t) + \hat{U}^{t-1}(x^t) - U_{\boldsymbol{\mathcal{D}}}(x^t) \\
&\leq \hat{U}^{t-1}(x^*) - \hat{U}^{t-1}(x^t) + 2 \cdot \mathcal{O}\left( \sqrt{\frac{L \log(t-1)}{t-1}} + \sqrt{\frac{L \log(nKA) + \log(\frac{1}{\delta})}{t-1}} \right) \\
&\leq 0 + 2 \cdot \mathcal{O}\left( \sqrt{\frac{L \log(t-1)}{t-1}} + \sqrt{\frac{L \log(nKA) + \log(\frac{1}{\delta})}{t-1}} \right),
\end{aligned}$$

where the last inequality follows from $\hat{U}^{t-1}(x^*) - \hat{U}^{t-1}(x^t) \leq 0$ because the algorithm selects the strategy $x^t$ that maximizes the empirical utility $\hat{U}^{t-1}(x)$. Then:

$$\begin{aligned}
\mathbb{E}[\mathrm{Reg}(T)] = \mathbb{E}\left[ \sum_{t=1}^{T} r(t) \right] \\
&\leq \sum_{t=1}^{T} (1 - \delta) \cdot \mathcal{O}\left( \sqrt{\frac{L \log t}{t}} + \sqrt{\frac{L \log(nKA) + \log(\frac{1}{\delta})}{t}} \right) + \delta T \qquad (5) \\
&\leq \mathcal{O}\left( \sqrt{TL \log T} + \sqrt{T\left( L \log(nKA) + \log(\frac{1}{\delta}) \right)} \right) + \delta T \qquad (6) \\
&\leq \mathcal{O}\left( \sqrt{TL(\log T + \log(nKA))} \right) \qquad \text{(Using inequality } \sqrt{a} + \sqrt{b} \leq \sqrt{2(a+b)} \text{)} \\
&= \mathcal{O}\left( \sqrt{TL \log(nKAT)} \right).
\end{aligned}$$

Equation (5) follows from the law of total expectation and the fact that the single-round regret is bounded by 1. Equation (6) follows from the known inequality $\sum_{t=1}^{T} \frac{1}{\sqrt{t}} \leq 2\sqrt{T}$. We set $\delta = \frac{1}{T}$.

**Runtime Analysis:** As for the computational complexity of this algorithm, note that Lemma 3.2 states that the number of non-empty best-response regions is $\mathcal{O}(n^L K^L A^{2L})$. As shown by Equation (3), we can compute the optimal strategy within each best-response region using a linear program with $L$ variables and at most $\mathrm{poly}(n^L, K^L, A^L, L, T)$ number of constraints. Further, evaluating each constraint and the objective function can also be accomplished in this time. Since each linear program can be solved in $\mathrm{poly}(n^L, K^L, A^L, L, T)$ time, and we run $\mathcal{O}(n^L K^L A^{2L})$ linear programs at each round, with at most $T$ rounds, Algorithm 1 runs $\mathrm{poly}((nKA)^L LT)$ time.

### B.3 PROOF OF THEOREM 4.2

*Proof.* Let $\boldsymbol{\mathcal{D}} = \prod_{i=1}^{n} \mathcal{D}_i$ denote the distribution over independent types. According to Claim B.1, the single-round regret $r(t) = U(x^*, \boldsymbol{\mathcal{D}}) - U(x^t, \boldsymbol{\mathcal{D}})$ satisfies

$$r(t) \leq \mathcal{O}(\delta(\boldsymbol{\mathcal{D}}, \hat{\boldsymbol{\mathcal{D}}}^{t-1})),$$

where $\hat{\boldsymbol{\mathcal{D}}}^{t-1} = \prod_{i=1}^{n} \hat{\mathcal{D}}_i^{t-1}$ is the product of the empirically computed marginal type distributions. We will use the following properties of Hellinger Distance (Definition B.2):

- (Guo et al., 2021) For any two distributions $\boldsymbol{\mathcal{D}}$ and $\hat{\boldsymbol{\mathcal{D}}}$,

$$H^2(\boldsymbol{\mathcal{D}}, \hat{\boldsymbol{\mathcal{D}}}) \leq \delta(\boldsymbol{\mathcal{D}}, \hat{\boldsymbol{\mathcal{D}}}) \leq \sqrt{2} H(\boldsymbol{\mathcal{D}}, \hat{\boldsymbol{\mathcal{D}}}). \tag{7}$$

- (Guo et al., 2021) If both $\boldsymbol{\mathcal{D}}$ and $\hat{\boldsymbol{\mathcal{D}}}$ are product distributions, i.e. $\boldsymbol{\mathcal{D}} = \prod_{i=1}^{n} \mathcal{D}_i$ and $\hat{\boldsymbol{\mathcal{D}}} = \prod_{i=1}^{n} \hat{\mathcal{D}}_i$, then:

$$H^2(\boldsymbol{\mathcal{D}}, \hat{\boldsymbol{\mathcal{D}}}) \leq \sum_{i=1}^{n} H^2\left(\mathcal{D}_i, \hat{\mathcal{D}}_i\right). \tag{8}$$

- (Canonne, 2020) For a distribution $\mathcal{D}$ with support size $K$, the empirical distribution $\hat{\mathcal{D}}^t$ over $t$ samples from $\mathcal{D}$ satisfies:

$$\mathbb{E}[H^2(\mathcal{D}, \hat{\mathcal{D}}^t)] \leq \frac{K}{2t}. \tag{9}$$

We now upper bound the single-round regret $r(t+1)$ in expectation:

$$
\begin{aligned}
\mathbb{E}[r(t+1)] &\leq \mathcal{O}\big(\mathbb{E}\big[\delta(\boldsymbol{\mathcal{D}}, \hat{\boldsymbol{\mathcal{D}}}^t)\big]\big) & \\
&\leq \mathcal{O}\big(\mathbb{E}\big[H(\boldsymbol{\mathcal{D}}, \hat{\boldsymbol{\mathcal{D}}}^t)\big]\big) & \text{by (7)} \\
&\leq \mathcal{O}\big(\sqrt{\mathbb{E}\big[H^2(\boldsymbol{\mathcal{D}}, \hat{\boldsymbol{\mathcal{D}}}^t)\big]}\big) & \text{because } \mathbb{E}[X^2] \geq (\mathbb{E}[X])^2 \\
&\leq \mathcal{O}\left(\sqrt{\mathbb{E}\Big[\sum_{i=1}^{n} H^2(\mathcal{D}_i, \hat{\mathcal{D}}_i^t)\Big]}\right) & \text{by (8)} \\
&\leq \mathcal{O}\left(\sqrt{\frac{nK}{2t}}\right). & \text{by (9)}
\end{aligned}
$$

Using the inequality $\sum_{t=1}^{T} \frac{1}{\sqrt{t}} \leq 2\sqrt{T}$, we obtain

$$\mathbb{E}[\mathrm{Reg}(T)] = \sum_{t=1}^{T} \mathbb{E}[r(t)] \leq \sum_{t=1}^{T} \mathcal{O}\left(\sqrt{\frac{nK}{t}}\right) \leq \mathcal{O}(\sqrt{nKT}).$$

$\square$

## B.4 $\Omega(\sqrt{\min\{L,K\}T})$ Lower Bound in the Single-Follower Case

In this section, we prove a lower bound of $\Omega(\sqrt{\min\{L,K\}T})$ on the expected regret of any algorithm in the case of a single-follower ($n = 1$), formalized in Theorem B.2.

**Theorem B.2.** *For single-follower Bayesian Stackelberg games where the follower has $K$ types and the leader has $L$ actions, the expected regret of any type-feedback online learning algorithm is at least $\Omega(\sqrt{\min\{L,K\}T})$.*

At a high level, the proof Theorem B.2 is a reduction from the *distribution learning problem*. Without loss of generality, assume that $\min\{K, L\} = 2c$ is an even number. Further assume that $K = L = 2c$.[5] The single follower has $K = 2c$ types, with type space $\Theta = \{\pm 1, \pm 2, ..., \pm c\}$. Consider a class $\mathcal{C}$ of distributions over $\Theta$ defined as follows:

**Definition B.4** (Class of Distributions $\mathcal{C}$). *A distribution $\mathcal{D} = \mathcal{D}_{\boldsymbol{\sigma}} \in \mathcal{C}$ is specified by a vector $\boldsymbol{\sigma} = (\sigma_1, \ldots, \sigma_c) \in \{\pm 1\}^c$. For each $j = 1, \ldots, c$,*

$$\mathcal{D}_{\boldsymbol{\sigma}}(+j) = \frac{1}{2c}(1 + \sigma_j \epsilon), \quad \mathcal{D}_{\boldsymbol{\sigma}}(-j) = \frac{1}{2c}(1 - \sigma_j \epsilon). \tag{10}$$

*for some $\epsilon > 0$. Note that $\mathcal{D}_{\boldsymbol{\sigma}}(+j) > \mathcal{D}_{\boldsymbol{\sigma}}(-j)$ if and only if $\sigma_j = +1$. The class $\mathcal{C}$ consists of $2^c$ distributions.*

In the distribution learning problem, given $t$ samples $\theta^1, ..., \theta^t$ from an unknown distribution $\mathcal{D} \in \mathcal{C}$, the goal is to construct an estimator $\hat{\mathcal{D}}$ specified by a vector $\hat{\sigma} \in \{\pm 1\}^c$ such that the expected total variation distance (Definition B.1) satisfies $\mathbb{E}[\delta(\mathcal{D}, \hat{\mathcal{D}})] \leq \mathcal{O}(\epsilon)$. It is known that solving this problem requires at least $\Omega(\frac{2c}{\epsilon^2})$ samples.

**Theorem B.3** (e.g., (Lee & Chen, 2020; Diakonikolas & Kontonis, 2019)). *When $\mathcal{D}$ is uniformly sampled from the class $\mathcal{C}$, any algorithm that constructs estimator $\hat{\mathcal{D}}$ using $t$ samples from $\mathcal{D}$ has expected error at least $\mathbb{E}[\delta(\hat{\mathcal{D}}, \mathcal{D})] \geq \Omega(\epsilon)$ if $t \leq \mathcal{O}(\frac{2c}{\epsilon^2})$.*

---

[5]If $K > 2c$, we can let the additional types to have probability 0. If $L > 2c$, we can let the additional actions of the leader to have very low utility.

---

**Reduction from distribution learning to single-follower Bayesian Stackelberg game**

---

**Distribution learning instance**: An unknown distribution $\mathcal{D} \in \mathcal{C}$.

**Bayesian Stackelberg game instance**: A single follower with type space $\Theta = \{\pm 1, \pm 2, \ldots, \pm c\}$ and an unknown type distribution $\mathcal{D}$. The follower has binary action set $\mathcal{A} = \{\text{Good}, \text{Bad}\}$. The leader has action set $\mathcal{L} = \Theta = \{\pm 1, \pm 2, \ldots, \pm c\}$. The utility functions of the two players are:

- Follower's utility function:

$$v(\ell, a, \theta) = \begin{cases} 1 & \text{if } \theta = +j, \ell = +j, a = \text{Good} \\ 1 & \text{if } \theta = +j, \ell = -j, a = \text{Bad} \\ 1 & \text{if } \theta = -j, \ell = -j, a = \text{Good} \\ 1 & \text{if } \theta = -j, \ell = +j, a = \text{Bad} \\ 0 & \text{otherwise.} \end{cases} \quad (11)$$

- Leader's utility function: For any action $\ell \in \mathcal{L}$,

$$u(\ell, \text{Good}) = 1, \quad u(\ell, \text{Bad}) = 0. \quad (12)$$

Note that for any mixed strategy $x$, $u(x, \text{Good}) = 1$ and $u(x, \text{Bad}) = 0$.

**Reduction:**
Given an online learning algorithm Alg for Bayesian Stackelberg game with type feedback, we use it to construct an online learning algorithm for the distribution learning problem as follows: At each round $t = 1, \ldots, T$,

1. Receive the leader's mixed strategy $x^t$ from Alg.

2. Construct an estimated distribution $\hat{\mathcal{D}}_{x^t} = \mathcal{D}_{\boldsymbol{\sigma}(x^t)} \in \mathcal{C}$ based on vector $\boldsymbol{\sigma}(x^t)$ defined as follows:
$$\sigma_j(x^t) = \begin{cases} +1, & \text{if } x^t(+j) \geq x^t(-j) \\ -1, & \text{if } x^t(+j) < x^t(-j). \end{cases} \quad (13)$$

3. Observe sample $\theta^t \sim \mathcal{D}$ and feed $\theta^t$ to Alg.

---

**Lemma B.2** (Follower's Best Response). *Given the leader's mixed strategy $x \in \Delta(\Theta)$, for each $j \in \{1, \ldots, c\}$, the best-response function of a follower with type $+j$ or $-j$ is:*

$$\mathrm{br}(+j, x) = \begin{cases} \text{Good}, & \text{if } x(+j) \geq x(-j), \\ \text{Bad}, & \text{if } x(+j) < x(-j), \end{cases}$$

$$\mathrm{br}(-j, x) = \begin{cases} \text{Good}, & \text{if } x(+j) < x(-j), \\ \text{Bad}, & \text{if } x(+j) \geq x(-j). \end{cases}$$

*Proof.* For a follower with type $+j$, their utility for choosing action Good is given by

$$v(x, \text{Good}, +j) = \mathbb{E}_{\ell \sim x}[v(\ell, \text{Good}, +j)] = \sum_{\ell \in \mathcal{L}} x(\ell) v(\ell, \text{Good}, +j) = x(+j).$$

Similarly, their utility for choosing action Bad is:

$$v(x, \text{Bad}, +j) = x(-j).$$

Thus, by definition, the follower with type $+j$ best responds with Good if $x(+j) \geq x(-j)$.

Likewise, for a follower with type $-j$,

$$v(x, \text{Good}, -j) = x(-j),$$
$$v(x, \text{Bad}, -j) = x(+j).$$

Thus, a follower with type $-j$ best responds with Bad if $x(+j) \geq x(-j)$, Good otherwise. $\qquad\square$

We define $U(x, \mathcal{D})$ as the expected utility of the leader when using mixed strategy $x$ under the type distribution $\mathcal{D}$. By Lemma B.2, we have

$$U(x, \mathcal{D}) = \sum_{\theta \in \Theta} \mathcal{D}(\theta) \, u\big(x, \mathrm{br}(\theta, x)\big)$$

$$= \sum_{j=1}^{c} \Big[ \mathcal{D}(+j) \, u\big(x, \mathrm{br}(+j, x)\big) + \mathcal{D}(-j) \, u\big(x, \mathrm{br}(-j, x)\big) \Big]$$

$$= \sum_{j=1}^{c} \Big( \frac{1 + \sigma_j \epsilon}{2c} \mathbb{1}\left[x(+j) \geq x(-j)\right] + \frac{1 - \sigma_j \epsilon}{2c} \mathbb{1}\left[x(+j) < x(-j)\right] \Big). \qquad (14)$$

**Definition B.5** (Disagreement Function). *The disagreement function* $\mathrm{Disagree}(x, \mathcal{D})$ *is the number of $j \in \{1, \ldots, c\}$ where the indicators $\mathbb{1}\left[x(+j) \geq x(-j)\right]$ and $\mathbb{1}\left[\mathcal{D}(+j) \geq \mathcal{D}(-j)\right]$ differ:*

$$\mathrm{Disagree}(x, \mathcal{D}) = \sum_{j=1}^{c} \mathbb{1}\Big[ \mathbb{1}\left[x(+j) \geq x(-j)\right] \neq \mathbb{1}\left[\mathcal{D}(+j) \geq \mathcal{D}(-j)\right] \Big]$$

$$= \sum_{j=1}^{c} \mathbb{1}\Big[ \mathbb{1}\left[x(+j) \geq x(-j)\right] \neq \mathbb{1}\left[\sigma_j = +1\right] \Big].$$

**Lemma B.3.** $U(\mathcal{D}, \mathcal{D}) - U(x, \mathcal{D}) = \frac{\epsilon}{c} \cdot \mathrm{Disagree}(x, \mathcal{D})$. *In particular, the optimal strategy for the leader is $x^* = \mathcal{D}$.*

*Proof.*

$$U(\mathcal{D}, \mathcal{D}) - U(x, \mathcal{D}) = \sum_{j=1}^{c} \Bigg( \frac{1 + \sigma_j \epsilon}{2c} \Big( \mathbb{1}\left[\mathcal{D}(+j) \geq \mathcal{D}(-j)\right] - \mathbb{1}\left[x(+j) \geq x(-j)\right] \Big)$$

$$+ \frac{1 - \sigma_j \epsilon}{2c} \Big( \mathbb{1}\left[\mathcal{D}(+j) < \mathcal{D}(-j)\right] - \mathbb{1}\left[x(+j) < x(-j)\right] \Big) \Bigg).$$

For each term in the summation where $\mathcal{D}(+j) \geq \mathcal{D}(-j)$ and $x(+j) \geq x(-j)$ agree, the term evaluates to $0$. When they disagree, there are two possible cases:

1. $\mathcal{D}(+j) \geq \mathcal{D}(-j)$ but $x(+j) < x(-j)$. Since $\mathcal{D}(+j) \geq \mathcal{D}(-j)$ implies $\sigma_j = +1$, the term simplifies to

$$\frac{1+\epsilon}{2c} - \frac{1-\epsilon}{2c} = \frac{\epsilon}{c}.$$

2. $\mathcal{D}(+j) < \mathcal{D}(-j)$ but $x(+j) \geq x(-j)$. Here, $\sigma_j = -1$, so the term simplifies to

$$-\frac{1-\epsilon}{2c} + \frac{1+\epsilon}{2c} = \frac{\epsilon}{c}.$$

Thus, we conclude that

$$U(\mathcal{D}, \mathcal{D}) - U(x, \mathcal{D}) = \frac{\epsilon}{c} \cdot \mathrm{Disagree}(x, \mathcal{D}).$$

$\square$

**Lemma B.4.** *Let $\hat{\mathcal{D}}_x$ be the estimated distribution constructed from $x$ according to Equation (13). The total variation distance between $\hat{\mathcal{D}}_x$ and $\mathcal{D}$ is given by*

$$\delta(\hat{\mathcal{D}}_x, \mathcal{D}) = \frac{\epsilon}{c} \cdot \mathrm{Disagree}(x, \mathcal{D}).$$

*Proof.* By definition,

$$\delta(\hat{\mathcal{D}}_x, \mathcal{D}) = \frac{1}{2} \sum_{j=1}^{c} \left( \left| \hat{\mathcal{D}}_x(+j) - \mathcal{D}(+j) \right| + \left| \hat{\mathcal{D}}_x(-j) - \mathcal{D}(-j) \right| \right).$$

If $x$ and $\mathcal{D}$ agree at $(+j)$ and $(-j)$, the corresponding term in the total variation sum is $0$. Consequently, we only need to consider the case when a disagreement occurs.

1. $x(+j) < x(-j)$ while $\mathcal{D}(+j) \geq \mathcal{D}(-j)$. In this case,

$$\hat{\mathcal{D}}_x(+j) = \frac{1-\epsilon}{2c}, \quad \hat{\mathcal{D}}_x(-j) = \frac{1+\epsilon}{2c}, \quad \mathcal{D}(+j) = \frac{1+\epsilon}{2c}, \quad \mathcal{D}(-j) = \frac{1-\epsilon}{2c}.$$

Hence,

$$\left| \hat{\mathcal{D}}_x(+j) - \mathcal{D}(+j) \right| + \left| \hat{\mathcal{D}}_x(-j) - \mathcal{D}(-j) \right| = \frac{\epsilon}{c} + \frac{\epsilon}{c} = \frac{2\epsilon}{c}.$$

2. $x(+j) \geq x(-j)$ while $\mathcal{D}(+j) < \mathcal{D}(-j)$. Similarly, we have

$$\hat{\mathcal{D}}_x(+j) = \frac{1+\epsilon}{2c}, \quad \hat{\mathcal{D}}_x(-j) = \frac{1-\epsilon}{2c}, \quad \mathcal{D}(+j) = \frac{1-\epsilon}{2c}, \quad \mathcal{D}(-j) = \frac{1+\epsilon}{2c}.$$

Again, we have

$$\left| \hat{\mathcal{D}}_x(+j) - \mathcal{D}(+j) \right| + \left| \hat{\mathcal{D}}_x(-j) - \mathcal{D}(-j) \right| = \frac{\epsilon}{c} + \frac{\epsilon}{c} = \frac{2\epsilon}{c}.$$

Thus, it follows that

$$\delta(\hat{\mathcal{D}}_x, \mathcal{D}) = \frac{\epsilon}{c} \cdot \mathrm{Disagree}(x, \mathcal{D}).$$

$\square$

From Lemma B.3 and Lemma B.4,

$$U(\mathcal{D}, \mathcal{D}) - U(x, \mathcal{D}) = \frac{\epsilon}{c} \cdot \mathrm{Disagree}(x, \mathcal{D}) = \delta(\hat{\mathcal{D}}_x, \mathcal{D}). \tag{15}$$

Consider the regret of the online learning algorithm Alg for the Bayesian Stackelberg game, where the algorithm outputs $x^t$ at round $t$. By Equation (15) and Theorem B.3, the expected regret at round $t \leq \mathcal{O}(\frac{2c}{\epsilon^2})$ is at least

$$\mathbb{E}[U(\mathcal{D}, \mathcal{D}) - U(x^t, \mathcal{D})] = \mathbb{E}[\delta(\hat{\mathcal{D}}_{x^t}, \mathcal{D})] \geq \Omega(\epsilon).$$

Thus, the expected regret over $T$ rounds is at least:

$$\mathbb{E}[\text{Reg}(T)] \;=\; \sum_{t=1}^{T} \mathbb{E}[U(\mathcal{D}, \mathcal{D}) - U(x^t, \mathcal{D})] \;\geq\; \min\left\{T, \; \mathcal{O}\left(\frac{2c}{\epsilon^2}\right)\right\} \cdot \Omega(\epsilon)$$

$$\geq\; \Omega(\sqrt{2cT}) \;=\; \Omega(\sqrt{\min\{K, L\}T})$$

where we choose $\epsilon = \sqrt{\frac{2c}{T}}$.

## B.5 $\Omega(\sqrt{\min\{L, nK\}T})$ LOWER BOUND FOR THE MULTI-FOLLOWER CASE: PROOF OF THEOREM 4.3

We now prove a lower bound of $\Omega(\sqrt{\min\{L, nK\}T})$ on the expected regret of any online learning algorithms for Bayesian Stackelberg games with multiple followers. Without loss of generality, assume that $nK$ is an even integer, and assume that the number of leader actions $L \geq nK$. We do a reduction from the single-follower problem to the multi-follower problem.

**Single-Follower Bayesian Stackelberg Game instance:** Consider the single-follower Bayesian Stackelberg game instance defined in Appendix B.4, but instead of a single follower with $K$ types, we change the instance so that the single follower has $nK$ types, indexed by $\Theta = \{(i, j) : i \in [n], j \in [K]\}$. Suppose the single follower's type distribution $\mathcal{D}$ belongs to the class $\mathcal{C}$ in Definition B.4 with support size $2c = nK$ (instead of $2c = K$). Note that for such a $\mathcal{D} \in \mathcal{C}$,

$$\sum_{i=1}^{n}\sum_{j=1}^{K} \mathcal{D}(i, j) = 1 \quad \text{and} \quad \forall i \in [n], \;\; \sum_{j=1}^{K} \mathcal{D}(i, j) = \frac{1}{n}.$$

The follower's utility function $v$ is given by (11), except that we now use $\theta = (i, j)$ to represent a type and $\ell = (i, j)$ to represent a leader's action. The leader's action set is $\mathcal{L} = \Theta$, with utility function $u$ given by (12).

**Multi-Follower Bayesian Stackelberg Game instance:** We reduce the single-follower game to an $n$-follower game defined below. Consider a Bayesian Stackelberg game with $n$ followers each with $K + 1$ types. The type distribution and the followers and leader's actions and utilities are defined below: (To distinguish the notations from the single-follower game, we use tilde notations $\tilde{\cdot}$)

- **Type distribution**: The followers' types are independently distributed according to distribution $\tilde{\mathcal{D}} = \prod_{i=1}^{n} \tilde{\mathcal{D}}_i$ where the probability that follower $i \in [n]$ has type $j$ is:

$$\tilde{\mathcal{D}}_i(j) = \begin{cases} 1 - \frac{1}{100n} & \text{if } j = 0, \\ \frac{1}{100}\mathcal{D}(i, j) & \text{if } j = 1, \dots, K. \end{cases}$$

- **Followers' actions and utilities**: Each follower has 3 actions $\tilde{\mathcal{A}} = \{\text{Good}, \text{Bad}, a_0\}$. The utility of a follower $i$ with type $j \neq 0$ is equal to the utility of the single follower with type $(i, j)$. Utilities for type $j = 0$ and action $a_0$ are specially defined:

$$\tilde{v}_i(\ell, a, \theta_i = j) = \begin{cases} v(\ell, a, (i, j)) & \text{if } \theta_i \neq 0 \text{ and } a \neq a_0 \\ -1 & \text{if } \theta_i \neq 0 \text{ and } a = a_0, \\ 1 & \text{if } \theta_i = 0 \text{ and } a = a_0, \\ -1 & \text{if } \theta_i = 0 \text{ and } a \neq a_0. \end{cases}$$

Note that the best-response action of a follower with type $0$ is always $a_0$, regardless of the leader's strategy.

- **Leader's actions and utilities:** The leader has the same action set as the single-follower game: $\mathcal{L} = \Theta = \{(i, j) : i \in [n], j \in [K]\}$. For any leader action $\ell \in \mathcal{L}$,

$$\tilde{u}(\ell, \boldsymbol{a}) = \begin{cases} 1 & \text{if } n - 1 \text{ followers choose } a_0 \text{ and one plays Good}, \\ 0 & \text{otherwise}. \end{cases}$$

---

**Reduction from Single-Follower Bayesian Stackelberg Game to Multi-Follower Game**

---

Given an online learning algorithm $\mathrm{Alg}$ for the $n$-follower problem, we construct an online learning algorithm for the single-follower problem as follows:

At each round $t = 1, \ldots, T$:

- Obtain a strategy $x^t \in \Delta(\mathcal{L})$ from algorithm $\mathrm{Alg}$. Output $x^t$.
- Receive a sample of the single follower's type $\theta^t = (i^t, j^t) \sim \mathcal{D}$.
- For every follower $i \in [n]$, we construct their type $\theta_i^t$ in the following way: Independently flip a coin that lands on head with probability $1 - \frac{1}{100n}$. If it lands on head, set the follower type $\theta_i^t$ to 0. If it lands on tail, we select the most recent sample of the form $(i^s = i, j^s)$ from the history $\{(i^s, j^s)\}_{s=1}^t$, and set the follower's type $\theta_i^t$ to $j^s$. Each sample can only be used once. If there are insufficient samples, we halt the algorithm.
- Provide the constructed types $(\theta_1^t, \ldots, \theta_n^t)$ to algorithm $\mathrm{Alg}$.

---

In the above reduction process, if we always have sufficient samples in the third step at each round, then the distribution of samples $(\theta_1^t, \ldots, \theta^t)$ provided to algorithm $\mathrm{Alg}$ is equal to the type distribution $\tilde{\mathcal{D}} = \prod_{i=1}^n \tilde{\mathcal{D}}_i$ of the $n$-follower game. Thus, from algorithm $\mathrm{Alg}$'s perspective, it is solving the $n$-follower game with unknown type distribution $\tilde{\mathcal{D}}$. We then argue that we have sufficient samples with high probability. Let $H_i^t$ be the number of available samples in the history that we can use to set follower $i$'s type at round $t$, and let $N_i^t$ be the number of samples that we actually need. Define

$$1 - \delta(t) = \Pr\left(\forall i \in [n], H_i^t \geq N_i^t\right),$$

which is the probability that we have sufficient samples at round $t$.

**Claim B.2.** $\delta(t) \leq 2n \exp\left(-\frac{t^2\left(\frac{1}{100n} - \frac{1}{n}\right)^2}{2t}\right).$

*Proof.* Note that $H_i^t$ and $N_i^t$ are Binomial random variables: $H_i^t \sim \mathrm{Bin}(t, \frac{1}{n})$, $N_i^t \sim \mathrm{Bin}(t, \frac{1}{100n})$. So, by union bound and Hoeffding's inequality:

$$\Pr\left(\exists i \in [n], H_i^t < N_i^t\right) \leq n \Pr\left(H_i^t < N_i^t\right) \leq 2n \exp\left(-\frac{t^2\left(\frac{1}{100n} - \frac{1}{n}\right)^2}{2t}\right).$$

$\square$

Let $\tilde{U}(x)$ be the leader's expected utility in the $n$-follower game (on type distribution $\tilde{\mathcal{D}}$) and $U(x)$ be the leader's utility in the single-follower game (on type distribution $\mathcal{D}$). We note that, given any strategy $x \in \Delta(\mathcal{L})$ of the leader, the best-response action of follower $i$ with type $\theta_i = j \neq 0$ (in the $n$-follower game) is equal to the best-response action of the single follower with type $(i, j)$, namely, $\mathrm{br}_i(j, x) = \mathrm{br}((i, j), x)$. Thus,

$$
\begin{aligned}
\tilde{U}(x) &= \Pr[\text{exactly one follower has a non-0 type}] \\
&\quad \cdot \mathbb{E}[\text{leader's utility} \mid \text{exactly one follower has a non-0 type}] + 0 \\
&= \left(1 - \frac{1}{100n}\right)^{n-1} \sum_{i=1}^n \sum_{j=1}^K \frac{1}{100} \mathcal{D}(i, j) \mathbb{1}\left[\mathrm{br}_i(j, x) = \text{Good}\right] \\
&= \left(1 - \frac{1}{100n}\right)^{n-1} \sum_{i=1}^n \sum_{j=1}^K \frac{1}{100} \mathcal{D}(i, j) \mathbb{1}\left[\mathrm{br}((i, j), x) = \text{Good}\right] \\
&= \frac{1}{100}\left(1 - \frac{1}{100n}\right)^{n-1} U(x) \\
&\approx \frac{1}{100} e^{-\frac{1}{100}} U(x).
\end{aligned}
$$

Define $C = \frac{1}{100}\left(1 - \frac{1}{100n}\right)^{n-1}$. Let $\tilde{r}(t) = \tilde{U}(x^*) - \tilde{U}(x^t)$ denote the per-round regret of the online learning algorithm $\mathrm{Alg}$ for the $n$-follower game. Let $r(t) = U(x^*) - U(x^t)$ denote the per-round

regret of the single-follower algorithm constructed by the above reduction. Consider the expected total regret in the $n$-follower game:

$$
\begin{aligned}
\mathbb{E}[\tilde{\mathrm{Reg}}(T)] &= \sum_{t=1}^{T} \mathbb{E}[\tilde{r}(t)] \\
&\geq \sum_{t=1}^{T} \Big( (1 - \delta(t)) \cdot \mathbb{E}[\tilde{r}(t)] \ - \ \delta(t) \cdot 1 \Big) \\
&= \sum_{t=1}^{T} \big(1 - \delta(t)\big) \cdot C \cdot \mathbb{E}[r(t)] \ - \ \sum_{t=1}^{T} \delta(t) \\
&\geq C \cdot \sum_{t=1}^{T} \mathbb{E}[r(t)] \ - \ \sum_{t=1}^{T} \delta(t) \cdot C \cdot 1 \ - \ \sum_{t=1}^{T} \delta(t) \\
&= C \cdot \mathbb{E}[\mathrm{Reg}(T)] \ - \ (C+1) \sum_{t=1}^{T} \delta(t).
\end{aligned}
$$

Now, we bound $\sum_{t=1}^{T} \delta(t)$. Consider a threshold $\tau$ such that for all $t \geq \tau$, we have $\delta(t) \leq \frac{1}{T^2}$. To find $\tau$, we solve

$$
2n \exp\Big( -\frac{(1/100n - 1/n)^2}{2}\tau \Big) \leq \frac{1}{T^2}.
$$

Rearranging, we choose $\tau$ such that

$$
\tau \geq \frac{\ln(2nT^2)}{C_\tau}
$$

where $C_\tau = (\frac{1}{100n} - \frac{1}{n})^2$. If $t \leq \tau$, we bound $\delta(t) \leq 1$. For $t > \tau$, we use the bound $\delta(t) \leq \frac{1}{T^2}$. Now, summing over all $t$,

$$
\sum_{t=1}^{T} \delta(t) \ \leq \ \tau + (T - \tau)\frac{1}{T^2} \ = \ \frac{\ln\big(2nT^2\big)}{C_\tau} + \frac{T - \tau}{T^2} \ \leq \ \mathcal{O}\Big(\frac{\ln T}{C_\tau} + \frac{1}{T}\Big) \ = \ \mathcal{O}(\log T).
$$

Then,

$$
\mathbb{E}[\tilde{\mathrm{Reg}}(T)] \ \geq \ C \cdot \mathbb{E}[\mathrm{Reg}(T)] - \mathcal{O}(\log T).
$$

The regret $\mathbb{E}[\mathrm{Reg}(T)]$ for a single-follower game where the follower has $nK$ types and the leader has $L = nK$ actions is at least $\Omega(\sqrt{nKT})$ by Theorem B.2. Thus, we obtain

$$
\mathbb{E}[\tilde{\mathrm{Reg}}(T)] \ \geq \ C \cdot \Omega(\sqrt{nKT}) - \mathcal{O}(\log T) \ = \ \Omega(\sqrt{nKT}),
$$

which is also $\Omega(\sqrt{\min\{L, nK\}T})$ because $L = nK$.

## C  APPENDIX FOR SECTION 5

### C.1  $\mathcal{O}(K^n\sqrt{T}\log T)$-REGRET ALGORITHM AND THE PROOF OF THEOREM 5.1

We show that the online learning problem for a Bayesian Stackelberg game with action feedback can be solved with $\mathcal{O}(K^n\sqrt{T}\log T)$ regret, by using a technique developed by Bernasconi et al. (2023).

Bernasconi et al. (2023) showed that the online learning problem for a linear program with unknown objective parameter can be reduced to a linear bandit problem. We first show that the Bayesian Stackelberg game (which is not a linear program as defined in Definition 2.2) can be reformulated as a linear program. Then, we use Bernasconi et al. (2023)'s reduction to reduce the linear program formulation of online Bayesian Stackelberg game to a linear bandit problem. A difference between our work and Bernasconi et al. (2023) is that, while they consider an adversarial online learning setting, we consider a stochastic online learning setting. Directly applying Bernasconi et al. (2023)'s result will lead to an $\widetilde{\mathcal{O}}(K^{\frac{3n}{2}}\sqrt{T})$ regret bound. Instead, we apply the OFUL algorithm for stochastic linear bandit (Abbasi-yadkori et al., 2011) to obtain a better regret bound of $\widetilde{\mathcal{O}}(K^n\sqrt{T})$.

**Step 1: Reformulate Bayesian Stackelberg game as a linear program.** First, we reformulate the Bayesian Stackelberg game optimization problem $\max_{x \in \Delta(\mathcal{L})} U_{\mathcal{D}}(x)$ (Definition 2.2), which is a nonlinear program by definition, into a linear program. Let variable $x$ represent a joint distribution over best-response function $W \in \mathcal{A}^{nK}$ and the leader's actions $\mathcal{L}$. Specifically,

$$x = (x(W, \ell))_{W \in \mathcal{A}^{nK}, \ell \in \mathcal{L}} \in \mathbb{R}^{A^{nK} \times L}, \quad \text{where} \sum_{W \in \mathcal{A}^{nK}, \ell \in L} x(W, \ell) = 1, \text{ and } x(W, \ell) \geq 0.$$

Alternatively, $x$ can be viewed as an $A^{nK} \times L$-dimensional matrix, where $W$ indexes the row and $\ell$ indexes the columns. We maximize the following objective (which is linear in $x$):

$$\max_x U(x) = \sum_{W \in \mathcal{A}^{nK}} \sum_{\ell \in \mathcal{L}} \sum_{\boldsymbol{\theta} \in \Theta^n} \mathcal{D}(\boldsymbol{\theta}) x(W, \ell) u(\ell, W(\boldsymbol{\theta})), \tag{16}$$

subject to the Incentive Compatibility (IC) constraint, meaning that the followers' best-response actions are consistent with $W$: $\forall W \in \mathcal{A}^{nK}, \forall i \in [n], \forall \theta_i \in \Theta, \forall a_i \in \mathcal{A}$,

$$\sum_{\ell \in \mathcal{L}} x(W, \ell) \Big( v_i(\ell, w_i(\theta_i), \theta_i) - v_i(\ell, a_i, \theta_i) \Big) \geq 0. \tag{17}$$

**Lemma C.1.** *With known distribution $\mathcal{D}$, the Bayesian Stackelberg game can be solved by the linear program (16)(17) in the following sense: there exists a solution $x$ to (16)(17) with only one non-zero row $x(W^*, \cdot)$, and this row $x(W^*, \cdot) \in \mathbb{R}^L$ is a solution to $\max_{x \in \Delta(\mathcal{L})} U_{\mathcal{D}}(x)$.*

*Proof.* First, we prove that the linear program (16)(17) contains an optimal solution with only one non-zero row. Suppose an optimal solution $x$ has two non-zero rows $W_1, W_2$:

$$\sum_{\ell \in L} x(W_1, \ell) = p_1 > 0, \quad \sum_{\ell \in L} x(W_2, \ell) = p_2 > 0.$$

Consider the conditional expected utility of these two rows. Because, when conditioned on row $i$, the conditional probability of playing action $\ell$ is $\frac{x(W, \ell)}{p_1}$, we have:

$$u_1 = \frac{1}{p_1} \sum_{\ell \in \mathcal{L}} \sum_{\boldsymbol{\theta} \in \Theta^n} x(W_1, \ell) \mathcal{D}(\boldsymbol{\theta}) u(\ell, W_1(\boldsymbol{\theta})),$$

$$u_2 = \frac{1}{p_2} \sum_{\ell \in \mathcal{L}} \sum_{\boldsymbol{\theta} \in \Theta^n} x(W_2, \ell) \mathcal{D}(\boldsymbol{\theta}) u(\ell, W_1(\boldsymbol{\theta})).$$

Without loss of generality, assume $u_1 \geq u_2$. We construct a new solution $x'$ by transferring probability mass from row $W_2$ to row $W_1$. Specifically, $x'$ is defined as follows:

$$x'(W_1, \ell) = \frac{p_1 + p_2}{p_1} x(W_1, \ell), \qquad \forall \ell \in \mathcal{L}.$$
$$x'(W_2, \ell) = 0, \qquad \forall \ell \in \mathcal{L},$$
$$x'(W_j, \ell) = x(W_j, \ell), \qquad \forall \text{ other } W_j, \forall \ell \in \mathcal{L}.$$

It is straightforward to verify that $x'$ satisfies the IC constraint. Now, we show that the utility of $x'$ is weakly greater than the utility of $x$.

$$\begin{aligned}
U(x') &= \sum_{\ell \in \mathcal{L}} \sum_{\boldsymbol{\theta} \in \Theta^n} \frac{p_1 + p_2}{p_1} x(W_1, \ell) \mathcal{D}(\boldsymbol{\theta}) u(\ell, W_1(\boldsymbol{\theta})) \\
&\quad + \text{ utility from rows other than } \{W_1, W_2\} \\
&= (p_1 + p_2) u_1 + \text{ utility from rows other than } \{W_1, W_2\} \\
&\geq p_1 u_1 + p_2 u_2 + \text{ utility from rows other than } \{W_1, W_2\} \\
&= U(x).
\end{aligned}$$

Note that the $W_2$ row of $x'$ has become $0$. We can apply this construction iteratively until only one row remains non-zero, without decreasing utility, thus obtaining an optimal solution with only one non-zero row.

Let $x^*$ be an optimal solution to the linear program (16)(17) with only one non-zero row $W^*$. Let $x^*_{BS} = \max_{x \in \Delta(\mathcal{L})} U_{\mathcal{D}}(x)$ be an optimal solution for the Bayesian Stackelberg game. We prove that $U(x^*) = U_{\mathcal{D}}(x^*_{BS})$.

First, we prove $U_{\mathcal{D}}(x^*_{BS}) \le U(x^*)$. Let $W^*_{BS}$ be the best-response function corresponding to $x^*_{BS}$, i.e., $x^*_{BS} \in R(W^*_{BS})$. We construct a feasible solution $x$ to the linear program (16)(17) by setting the row indexed by $W^*_{BS}$ to $x^*_{BS}$ and assigning zero values to all other rows. By definition, $x$ satisfies the IC constraint, so it is a feasible solution. Moreover, $x^*_{BS}$ and $x$ achieve the same objective value $U_{\mathcal{D}}(x^*_{BS}) = U(x)$. By definition, $U(x)$ is weakly less than the optimal objective value $U(x^*)$ of the linear program, so $U_{\mathcal{D}}(x^*_{BS}) \le U(x^*)$.

Then, we prove $U(x^*) \le U_{\mathcal{D}}(x^*_{BS})$. Suppose the leader uses the strategy defined by the non-zero row of $x^*$, which is $x^*(W^*, \cdot) \in \Delta(\mathcal{L})$. By the IC constraint of the linear program, the best-response function of the followers is equal to $W^*$, so the expected utility of the leader is exactly equal to $U(x^*)$, which is $\le U_{\mathcal{D}}(x^*_{BS})$ because $x^*_{BS}$ is an optimal solution for the Bayesian Stackelberg game. $\square$

**Step 2: Reduce online Bayesian Stackelberg game to a linear bandit problem.** Based on the linear program formulation (16)(17), we then reduce the online Bayesian Stackelberg game problem to a linear bandit problem, using the technique in Bernasconi et al. (2023). Let $\mathcal{X} \subseteq \Delta(\mathcal{A}^{nK} \times \mathcal{L})$ be the set of feasible solutions to the linear program (16)(17). We define the loss of a strategy $x \in \mathcal{X}$ when the follower types are $\boldsymbol{\theta} \in [K]^n$ as:

$$L_{\boldsymbol{\theta}}(x) = - \sum_{W \in \mathcal{A}^{nK}} \sum_{\ell \in \mathcal{L}} x(W, \ell) u(\ell, W(\boldsymbol{\theta})).$$

We define a linear map $\phi : \mathcal{X} \to \mathbb{R}^{K^n}$ that maps a strategy $x \in \mathcal{X}$ to a vector in $\mathbb{R}^{K^n}$, representing the loss of the strategy for each type profile:

$$\phi(x) = \begin{pmatrix} L_{\boldsymbol{\theta}_1}(x) \\ \vdots \\ L_{\boldsymbol{\theta}_{K^n}}(x) \end{pmatrix} \in \mathbb{R}^{K^n}.$$

Its inverse, $\phi^\dagger : \mathbb{R}^{K^n} \to \mathcal{X}$ maps a loss vector back to a strategy. Let $\mathrm{co}\, \phi(\mathcal{X})$ denote the convex hull of the image set of $\phi$.

Let $\mathfrak{R}$ be a stochastic linear bandit algorithm with decision space $\mathrm{co}\, \phi(\mathcal{X}) \subseteq \mathbb{R}^{K^n}$. In particular, we let $\mathfrak{R}$ be the OFUL algorithm (Abbasi-yadkori et al., 2011). At each round, $\mathfrak{R}$ outputs a strategy $z^t \in \mathrm{co}\, \phi(\mathcal{X})$, and we invoke a Carathédory oracle to decompose $z^t$ into $K^n + 1$ elements from $\phi(\mathcal{X})$, forming a convex combination.[6] We then sample one of the elements $z^t_j$, and apply the inverse map $\phi^\dagger$ to obtain a strategy $x^t \in \mathcal{X}$ for the leader. After playing strategy $x^t$, we observe the utility $u^t = u(\ell^t, \boldsymbol{a}^t)$ and feed the utility feedback to $\mathfrak{R}$.

**Theorem C.1.** *The expected regret of Algorithm 5 is $O(K^n \sqrt{T} \log T)$.*

*Proof.* Because $x^t \in \mathcal{X} = \Delta(\mathcal{A}^{nK} \times \mathcal{L})$ is a feasible solutions to the linear program (16)(17), it satisfies the IC constraint. So, when the leader plays $x^t(W, \cdot)/p(W) \in \Delta(\mathcal{L})$, the followers (with types $\boldsymbol{\theta}$) will best respond according to the function $W(\boldsymbol{\theta})$. Thus, the leader's expected utility at round $t$ is

$$\sum_{W \in A^{nK}} p(W) \sum_{\ell \in \mathcal{L}} \frac{x^t(W, \ell)}{p(W)} \sum_{\boldsymbol{\theta} \in [K]^n} \mathcal{D}(\boldsymbol{\theta}) u(\ell, W(\boldsymbol{\theta})) = U(x^t) = -\mathbb{E}_{\boldsymbol{\theta} \sim \mathcal{D}}[L_{\boldsymbol{\theta}}(x^t)].$$

Then, the regret of Algorithm 5 in $T$ rounds can be expressed as

$$\mathrm{Reg}(T) = \sum_{t=1}^{T} \left( \mathbb{E}_{\boldsymbol{\theta} \sim \mathcal{D}}[L_{\boldsymbol{\theta}}(x^t)] - \mathbb{E}_{\boldsymbol{\theta} \sim \mathcal{D}}[L_{\boldsymbol{\theta}}(x^*)] \right),$$

---

[6]The Carathédory oracle is based on the well-known *Carathédory Theorem*.

---

**ALGORITHM 5:** Linear Bandit Algorithm for Bayesian Stackelberg Games

---

**Input :** A linear bandit algorithm $\mathfrak{R}$ over decision space $\mathrm{co}\,\phi(\mathcal{X})$, where $\mathfrak{R}.\mathrm{RECOMMEND}()$ returns an element in $\mathrm{co}\,\phi(\mathcal{X})$, and $\mathfrak{R}.\mathrm{OBSERVELOSS}$ takes the loss feedback.

1 **for** *each round $t$* **do**

2   Use $\mathfrak{R}.\mathrm{RECOMMEND}()$ to obtain $z^t \in \mathrm{co}\,\phi(\mathcal{X}) \subseteq \mathbb{R}^{K^n}$.

3   Call a Carathéodory oracle with input $(z^t, \phi(\mathcal{X}))$, which returns $K^n + 1$ elements $\{z_i^t, \lambda_i^t\}_{i \in [K^n+1]}$ such that:

$$z^t = \sum_{i=1}^{K^n+1} \lambda_i^t z_i^t, \quad \text{where} \quad \sum_{i=1}^{K^n+1} \lambda_i^t = 1.$$

4   Draw an index $j \in \{1, \ldots, K^n + 1\}$ with probabilities $\lambda_j^t$.

5   Compute $x^t \leftarrow \phi^\dagger(z_j^t)$. Note that $x^t \in \mathcal{X} = \Delta(\mathcal{A}^{nK} \times \mathcal{L})$ is a matrix.

6   Play $x^t$ in the following sense: sample a row $W \in \mathcal{A}^{nK}$ with probability $p(W) = \sum_{\ell \in \mathcal{L}} x^t(W, \ell)$, then play the mixed strategy $x^t(W, \cdot)/p(W) \in \Delta(\mathcal{L})$.

7   Observe the realized utility $u^t = u(\ell^t, \boldsymbol{a}^t)$.

8   Feed the loss to $\mathfrak{R}$ by calling $\mathfrak{R}.\mathrm{OBSERVELOSS}(-u^t)$.

9 We let $\mathfrak{R}$ be the OFUL algorithm (Abbasi-yadkori et al., 2011).

---

where $x^*$ is the optimal strategy in $\mathcal{X}$, which minimizes the expected loss (maximizes expected utility). Let $\mathrm{Reg}_{\mathfrak{R},\mathrm{co}\,\phi(\mathcal{X})}(T)$ be the expected regret of the linear bandit algorithm $\mathfrak{R}$ on decision space $\mathrm{co}\,\phi(\mathcal{X})$ in $T$ rounds. According to the Theorem 3.1 of Bernasconi et al. (2023),

$$\mathrm{Reg}(T) \le \mathrm{Reg}_{\mathfrak{R},\mathrm{co}\,\phi(\mathcal{X})}(T).$$

We let $\mathfrak{R}$ be the OFUL algorithm (Abbasi-yadkori et al., 2011). For any $z \in \mathrm{co}\,\phi(\mathcal{X}) \subseteq \mathbb{R}^{K^n}$, the stochastic loss of $z$ can be expressed as $L^t = \langle z, \boldsymbol{\mathcal{D}} \rangle + \eta^t$, with $|L^t| \le 1$, $\|\boldsymbol{\mathcal{D}}\|_2 \le \|\boldsymbol{\mathcal{D}}\|_1 = 1$, and $\eta^t$ being a bounded zero-mean noise. Then, from Abbasi-yadkori et al. (2011)'s Theorem 3, we have with probability at least $1 - \delta$,

$$\mathrm{Reg}_{\mathfrak{R},\mathrm{co}\,\phi(\mathcal{X})}(T) \le 4\sqrt{TK^n \log\left(\lambda + \frac{TL}{K^n}\right)} \cdot \left(\lambda^{1/2} + \sqrt{2\log\left(\frac{1}{\delta}\right) + K^n \log\left(1 + \frac{TL}{\lambda K^n}\right)}\right)$$

where $\lambda$ is a tunable parameter in the OFUL algorithm. By setting $\lambda = 1$ and $\delta = \frac{1}{T}$, we obtain

$$\mathbb{E}[\mathrm{Reg}(T)] \le (1-\delta) \cdot O(K^n \sqrt{T} \log T) + \delta \cdot T = \mathcal{O}(K^n \sqrt{T} \log T).$$

$\square$

## C.2 Proof of Lemma 5.1

We can express the leader's utility function as

$$u(x, \boldsymbol{a}) = \sum_{\ell \in \mathcal{L}} x(\ell) u(\ell, \boldsymbol{a}) = \langle x, u_{\boldsymbol{a}} \rangle$$

where vector $u_{\boldsymbol{a}} = (u(\ell, \boldsymbol{a}))_{\ell \in \mathcal{L}} \in \mathbb{R}^L$. Note that $u(x, \boldsymbol{a})$ is a linear function of $u_{\boldsymbol{a}}$. Consequently, the expected utility of a strategy $x \in R(W)$ on the true distribution $\boldsymbol{\mathcal{D}}$ is given by

$$U(x, R(W)) = \mathbb{E}_{\boldsymbol{a} \sim \mathcal{P}(\cdot|R(W))}[\langle x, u_{\boldsymbol{a}} \rangle].$$

Given samples $\boldsymbol{a}^1, ..., \boldsymbol{a}^N$, we can compute $u_{\boldsymbol{a}^1}, ..., u_{\boldsymbol{a}^N}$ because we know the utility function. By Lemma B.1, the pseudo-dimension of the family of linear functions $\{\langle x, \cdot \rangle \mid x \in R(W) \in \mathbb{R}^L\}$ is $L$. Applying Theorem B.1, with $N$ samples, we have

$$\Pr\left[\exists x \in R(W),\ \left|U(x, R(W)) - \hat{U}_N(x, R(W))\right| > \sqrt{\frac{2L \log 3N}{N}} + \sqrt{\frac{\log\frac{1}{\delta}}{2N}}\right] \le \delta.$$

Let $\delta = \frac{1}{T^4}$. Taking a union bound over all $N \in \{1, ..., T\}$ and all $W \in \mathcal{W}$, we obtain

$$\Pr\left[\exists W \in \mathcal{W}, \exists N \in [T], \exists x \in R(W), \left|U(x, R(W)) - \hat{U}_N(x, R(W))\right| > \sqrt{\frac{2L\log(3N)}{N}} + \sqrt{\frac{\log T^4}{2N}}\,\right]$$

$$\leq |\mathcal{W}|T\delta = \frac{|\mathcal{W}|T}{T^4} \leq \frac{1}{T^2}$$

(assuming $T \geq |\mathcal{W}|$). Thus, with probability at least $1 - \frac{1}{T^2}$, for every $W \in \mathcal{W}$, $N \in [T]$, and $x \in R(W)$, we have

$$\left|U(x, R(W)) - \hat{U}_N(x, R(W))\right| \leq \sqrt{\frac{4(L+1)\log(3T)}{t}}$$

using the inequality $\sqrt{a} + \sqrt{b} \leq \sqrt{2(a+b)}$.

## C.3 PROOF OF THEOREM 5.2

By Lemma 5.1, the event

$$C = \left[\forall W \in \mathcal{W}, \forall N \in [T], \forall x \in R(W), \left|U(x, R(W)) - \hat{U}_N(x, R(W))\right| \leq \sqrt{\frac{4(L+1)\log(3T)}{N}}\,\right]$$

happens with probability at least $1 - \frac{1}{T^2}$. Suppose $C$ happens. The regret at round $t$ is given by

$$r(t) = U(x^*, R(W^*)) - U(x^t, R(W^t)).$$

For any strategy $x \in R(W)$, we define the upper confidence bound of its utility as

$$\text{UCB}^t(x) = \hat{U}_{N^t(W)}(x, R(W)) + \sqrt{\frac{4(L+1)\log(3T)}{N^t(W)}}.$$

Since $C$ holds, it follows that
$$U(x^*, R(W^*)) \leq \text{UCB}^t(x^*).$$

Because the UCB algorithm chooses the strategy with the highest upper confidence bound at round $t$, we have $\text{UCB}^t(x^*) \leq \text{UCB}^t(x^t)$. Thus,

$$\begin{aligned}
r(t) &\leq \text{UCB}^t(x^*) - U(x^t, R(W^t)) \\
&\leq \text{UCB}^t(x^t) - U(x^t, R(W^t)) \\
&= \hat{U}_{N^t(W)}(x^t, R(W^t)) - U(x^t, R(W^t)) + \sqrt{\frac{4(L+1)\log(3T)}{N^t(W^t)}} \\
&\leq 2\sqrt{\frac{4(L+1)\log(3T)}{N^t(W^t)}}.
\end{aligned}$$

The total regret is at most

$$\begin{aligned}
\text{Reg}(T) &= \sum_{t=1}^{T} r(t) \\
&\leq 2\sum_{t=1}^{T} \sqrt{\frac{4(L+1)\log(3T)}{N^t(W^t)}} \\
&= 2\sum_{W \in \mathcal{W}} \sum_{m=1}^{N^T(W)} \sqrt{\frac{4(L+1)\log(3T)}{m}} \\
&\leq 8\sum_{W \in \mathcal{W}} \sqrt{N^T(W) \cdot (L+1) \cdot \log(3T)}
\end{aligned}$$

where we applied the inequality $\sum_{m=1}^{N} \sqrt{\frac{1}{m}} \leq 2\sqrt{N}$. By Jensen's inequality,

$$\frac{1}{|\mathcal{W}|} \sum_{W \in \mathcal{W}} \sqrt{N^T(W)} \leq \sqrt{\frac{1}{|\mathcal{W}|} \sum_{W \in \mathcal{W}} N^T(W)} = \sqrt{\frac{1}{|\mathcal{W}|} T}.$$

Thus,

$$\begin{aligned}
\text{Reg}(T) &\leq 8 \sum_{W \in \mathcal{W}} \sqrt{|\mathcal{W}|T} \cdot \sqrt{(L+1)\log(3T)} \\
&= \mathcal{O}\left(\sqrt{|\mathcal{W}|L \cdot T \log T}\right) \\
&= \mathcal{O}\left(\sqrt{n^L K^L A^{2L} L \cdot T \log T}\right)
\end{aligned}$$

where we used $|\mathcal{W}| = O(n^L K^L A^{2L})$ from Lemma 3.2.

Finally, considering the case where $C$ does not happen (which has probability at most $\frac{1}{T^2}$),

$$\mathbb{E}[\text{Reg}(T)] = \left(1 - \frac{1}{T^2}\right)\mathcal{O}\left(\sqrt{n^L K^L A^{2L} L \cdot T \log T}\right) + \frac{1}{T^2} \cdot T \leq \mathcal{O}\left(\sqrt{n^L K^L A^{2L} L \cdot T \log T}\right).$$

## D  SIMULATIONS

We empirically simulate and validate the results of the studied algorithms in both the type-feedback setting and action feedback setting. For the former, we consider the independent type setting to understand how much better, in practice, is Algorithm 2 (customized for independent types) as opposed to the general purpose Algorithm 1 (works for general type distributions). We consider an $(L = 2, K = 6, A = 2, n = 2)$ instance and simulate the results in Figure 2. As expected, the specialized algorithm does indeed outperform the general one.

For the action feedback case, we empirically compare our UCB-based Algorithm 3 with the linear bandit approach inspired by Bernasconi et al. (2023), Algorithm 5. We especially consider the small $n, L$ regime where our theory does not provide any concrete guidance. Shown in Figure 3, we consider an $(L = 2, K = 6, A = 2, n = 2)$ instance and observe the advantage of the UCB-based algorithm over the linear bandit one.

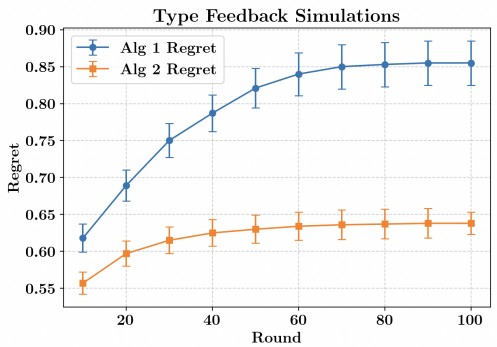
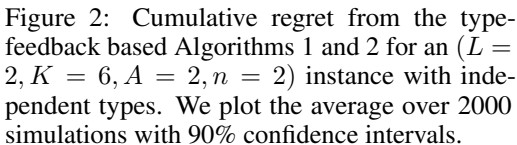
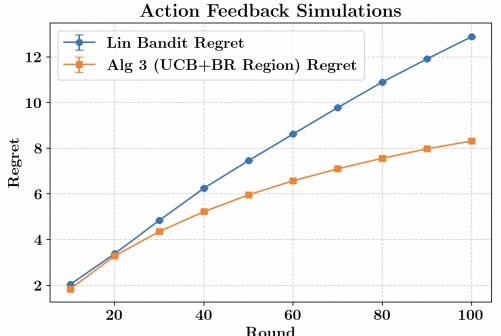

Figure 2: Cumulative regret from the type-feedback based Algorithms 1 and 2 for an $(L = 2, K = 6, A = 2, n = 2)$ instance with independent types. We plot the average over 2000 simulations with 90% confidence intervals.

Figure 3: Cumulative regret from Algorithm 5 (the Linear-Bandit approach inspired by Bernasconi et al. (2023)) and Algorithm 3 for an $(L = 2, K = 6, A = 2, n = 2)$ instance. We plot the average over 2000 simulations with 90% confidence intervals.

