# OpenReview forum: "Learning to Play Multi-Follower Bayesian Stackelberg Games"
_ICLR.cc/2026/Conference — ICLR 2026 Poster_

### Official Review · Reviewer_av7X · 2025-10-26

**Soundness:** 3
**Presentation:** 3
**Contribution:** 2
**Rating:** 6
**Confidence:** 3

**Summary:**

The paper studies online learning for multi-follower Bayesian Stackelberg games where a leader commits to a mixed strategy and $n$ followers (each with one of $K$ private types) best respond. The authors analyze two feedback models: type feedback and action feedback. The paper proposes a geometric decomposition of the leader’s simplex into best-response regions and algorithms with sublinear regret. For type feedback they give $\tilde{O}(\sqrt{\min\{L,nK\}}\,T)$ for independent types, $\tilde{O}(\sqrt{\min\{L,Kn\}}\,T)$ for general types, and a matching lower bound $\Omega(\sqrt{\min\{L,nK\}}\,T)$. For action feedback they propose a reduction to stochastic linear bandits with $O(Kn\sqrt{T}\log T)$ regret and a UCB  approach with $O(\sqrt{n^{L}K^{L}A^{2L}}\,L\,T\log T)$ regret.

**Strengths:**

The paper introduces a clean geometric decomposition of the leader’s strategy space into follower best-response regions and leverages this structure to derive regret guarantees for multi-follower Bayesian Stackelberg games. The resulting bounds avoid polynomial growth in the number of followers $n$, and the authors provide a lower bound that nearly matches the type-feedback upper bounds.

**Weaknesses:**

1. In the general single-leader-multi-follower setting, a follower’s utility typically depends on both the leader’s action and other followers’ actions (i.e., there are cross-follower externalities). The paper assumes each follower’s payoff depends only on their own action and type plus the leader’s action—which substantially simplifies the model and makes the extension from one to many followers more direct. While this assumption enables clean analysis, it limits the applicability of the results to settings without strategic interdependence among followers.
2. The analysis assumes followers play the exact best response in every round. This is a strong assumption for a learning setting: in many practical environments followers also learn, face estimation error, or act under noisy rewards, leading to approximate or stochastic best responses. The results would be more interesting with guarantees to $\varepsilon$-best responses or noise in followers’ payoff observations.
3. The reduction to stochastic linear bandits is not novel. Similar reductions have appeared in prior work [1].


[1] Nearly-optimal bandit learning in stackelberg games with side information. Balcan, Maria-Florina, et al.

**Questions:**

Could you elaborate more the statement: “In comparison, Conitzer & Sandholm (2006) prove that the optimal strategy is NP-hard to compute in BSGs with an asymptotically increasing $L$. We show this is polynomial-time solvable for a constant $L$.” (Line 274)
Could you cite the exact theorem/lemma in Conitzer & Sandholm, specify the precise problem and briefly contrast your assumptions with theirs to explain why constant $L$ yields polynomial time here.

---

> ### Author Response · Authors · 2025-11-23
>
> > Weakness about no externality:
>
> Without externality, even though the extension from one to many followers seems direct, the problem is still non-trivial. One might naturally expect the regret bound to grow at the exponential rate of $O(\sqrt{K^n T})$, because the support size of the joint type distribution is $K^n$. However, our work shows that the regret bounds can be significantly smaller. For example,
>
> (1) In the type feedback case with independent types, we achieve a polynomial regret $O(\sqrt{nKT})$ (Theorem 4.2).
>
> (2) In the action feedback case, our Algorithm 3, based on a combination of UCB and best response regions, achieve an $O(\sqrt{n^L K^L A^{2L} L T})$ regret, which is polynomial in $n$.
>
> In a model with externality, a leader strategy induces a simultaneous game among the followers. The corresponding equilibrium concept is Stackelberg-Nash equilibrium, where the followers best respond to each other and the leader.  However, computing the Nash equilibrium among the followers is PPAD-hard in the worst case.  So, for such a setting to be well motivated, we need to restrict attention to specific games where the Stackelberg-Nash equilibrium is computationally tractible.
>
> Putting aside the tractability of inter-follower Nash Equilibrium, some of our results in the type-feedback setting generalize: our $O(\sqrt{K^n T})$ result for general type distributions and $O(\sqrt{nKT})$ result for independent type distributions also apply to the setting with externality.  This is because externality changes _how the followers act_, but not their types. Our algorithms in the type-feedback setting efficiently learn the type distributions from samples, so also work in the presence of externality.  Our other results, building on best-response region characterization and depending on the independence of followers’ utility functions, do not generalize to the setting with externality.  We think externality is an interesting direction for future work, and added a discussion to the revised PDF.
>
>
> > (Weakness 2) Approximate or stochastic best response
>
> If followers do $\epsilon$-best responses with small $\epsilon$, then the leader can always perturb her strategy slightly to make a follower’s best responding action strictly better than any other actions by a margin of $\epsilon$, so the $\epsilon$-best responding behavior of the follower will be the same as exact best response.  In that case, our results are still applicable, except that the leader will suffer an $O(\epsilon)$ utility loss due to such perturbation.  Regarding stochastic best response, previous work (e.g., [Wu et al, NeurIPS 2022, Inverse Game Theory for Stackelberg Games: the Blessing of Bounded Rationality]) considered learning the follower’s utility function (instead of type distribution) in non-Bayesian Stackelberg games where the follower does quantal response (i.e., softmax response).  Learning the followers’ type distributions in Bayesian Stackelberg games with stochastic response might require very different techniques (than the techniques in previous work and our work) and is an interesting future research direction.
>
> > (Weakness 3)
>
> While the reduction to stochastic linear bandits is similar to prior work [Bernasconi et al (2023), Balcan et al (2025)], it is only a minor result in our action-feedback setting.  Our main result for the action-feedback setting is Algorithm 3, which achieves $O(\sqrt{n^L K^L A^{2L} LT \log T})$ regret and is better than the $O(K^n\sqrt{T} \log T)$ regret obtained by the stochastic linear bandit reduction, when $n$ is large.  Our Algorithm 3 combines the UCB principle and the best-response region characterization, which is a new approach to our knowledge.
>
>
> > Elaborate more on the statement: “In comparison, Conitzer & Sandholm (2006) prove that the optimal strategy is NP-hard to compute in BSGs with an asymptotically increasing . We show this is polynomial-time solvable for a constant L.”
>
> Conitzer & Sandholm (2006) consider Bayesian Stackelberg games with known distributions of follower types.  Their Theorem 7 shows that, even with a single leader and single follower, as the number of leader actions $L$ increases, computing the optimal leader strategy is NP-hard.  Our online learning problem, with unknown type distributions, is even harder than their problem, so is also NP-hard when $L$ increases.  In contrast, when treating $L$ as a constant, our geometric characterization and best-response region enumeration algorithm (Lemm 3.3) show that the optimal leader strategy can be computed in $polynomial(n^L, K^L, A^L, L)$ time (with known type distributions). This fixed-parameter tractability result is not given in [Conitzer & Sandholm, 2006] and we believe is of independent interest. We have added these clarifications to Line 283.

---

### Official Review · Reviewer_F5NQ · 2025-11-01

**Soundness:** 3
**Presentation:** 2
**Contribution:** 3
**Rating:** 6
**Confidence:** 4

**Summary:**

The paper studies multi-follower online Bayesian Stackelberg games. The authors study 2 types of feedback: type feedback, where at each turn the leader observes the tuple of realized types, and action feedback, in which the leader observes the individual actions. Clearly knowing the types implies knowing the actions.
Types are either drawn from a joint or product distribution.
Under type feedback the authors gives an algorithm which attains regret of $\sqrt{T\min(L,K^n)}$  under joint distribution, or, $\sqrt{T\min(L,Kn)}$ under product distribution, where $L$ is the number of actions of the followers, $K$ the number of possible types and $n$ the number of players. The algorithm revolves around
Under type feedback (under joint distribution) the algorithm is simply follow the leader. Interestingly, the analysis shows concentration around the empirical utility rather than under the correct type distribution. This aligns with the following section on action feedback, where working in the utility space is the correct trick to get around exponential dependencies.
Under type feedback and independent distribution, the algorithm simply estimated each distribution separately.
Under action feedback, the authors rely on a known reduction for a related problem of online Bayesian persuasion. The technical challenge here is to show that there exists a single LP that solves the problem.

**Strengths:**

This is a good paper that has non trivial contributions, even if maybe a bit lackluster from a technical side.

**Weaknesses:**

I see two main weaknesses:
1. Even if I find the overall presentation satisfactory, its quality degrades in some specific points. It is not clear where do you use the best response partitions. For example, the discussion under Lemma 4.1 seems to be very important, but right now it feels somewhat obscure and does not do a good job at explaining the crucial points. Another point in which the presentation is a bit confused is around line 409. It is not even clear if this part relies at all on the reduction of Bernasconi et al. or not. I think not, but better clarity would be nice. Overall, I think the paper suffers from having too many results compressed. I would suggest to the authors that maybe they defer some of the results to the appendix and devote more space to points that are now a bit compressed.
2. Why do you not consider adversarial types? I think your FTL approach would still be a good candidate for adversarial types (obviously by using FTPL instead). The reduction of Bernasconi for sure also works for adversarial types. So it is a bit strange that you do not consider this problem, which is the most studied in Online learning in Econ settings.

* Typo in statement of Lemma C.1

**Questions:**

Is the improvement from K^{3n/2} to K^n only due to the fact that you consider stochastic types rather than adversarial ones?

---

> ### Author Response · Authors · 2025-11-23
>
> > Clarifications 1: Where do we use best-response partition?
>
> The best response partition turns the problem into a discrete bandit problem, so we can prove concentration bounds separately for each region (Lemma 4.1) in the type feedback case, and apply UCB (an algorithm for a finite set of arms) to the finite set of best response regions in the action feedback case. We have added clarifications to Line 192-195.
>
> **We then provide more details regarding how Lemma 4.1 uses best response regions:**
>
> In the proof of Lemma 4.1, we partition the leader’s strategy space into $O(n^L K^L A^{2L})$ best-response regions.  By Lemma 3.1, the leader’s utility function is linear inside each region.  Because the pseudo-dimension of linear functions are $L$, we have with probability at least $1-\delta'$, the empirical utility $\hat U^t(x)$ approximates the true expected utility $U_D(x)$ with high accuracy for every strategy $x$ inside a best-response region.  Taking a union bound over all best-response regions, i.e., letting $\delta' = \delta / O(n^L K^L A^{2L})$, proves Lemma 4.1.  We added these clarifications after Lemma 4.1.
>
> > Clarification 2: Line 409 and relation to [Bernasconi et al. 2023]
>
> In the original PDF, below Line 409 is our second algorithm for the action-feedback setting – Algorithm 3. It does not rely on the reduction of Bernasconi et al (2023), as the reviewer correctly pointed out.  It uses a “concentration over best-response regions” idea that we developed in previous sections.  We have clarified this at the new Line 429.
>
> > (Weakness 1) Having too many results compressed.
>
> We’d like to provide a complete picture for the online Bayesian Stackelberg games problem. The regret upper bounds turn out to be of the form $min(A, B) \sqrt{T}$, requiring taking the better one between two algorithms.  This complication might be unavoidable according to our lower bound results.  But we do note that both of our main results for the type feedback (Theorem 4.1) and action feedback (Theorem 5.2) cases use the same technique: concentration over best-response regions.  These two results with this technique are the focus of our work. For other side results, we have indeed put the details in the appendix.
>
>
> > (Weakness 2) Why do you not consider adversarial types?
>
> First, we think the stochastic problem, even if it’s a special adversarial problem, is worthwhile to study specifically. For example, in the stochastic setting with independent follower types, we show that the regret can be significantly reduced to $O(\sqrt{T min(L, Kn)})$ with type feedback. That small regret is unlikely to be achievable in the adversarial setting.  We also show that, with action feedback, the $O(K^{3n/2}\sqrt{T})$ result for adversarial types from prior work can be improved to $O(K^n \sqrt{T})$ in the stochastic case.  We find these specific results for stochastic settings interesting by themselves.
>
> That said, we agree that the adversarial setting is also interesting. Regarding the reviewer’s suggestion of using Follower-the-Perturbed-Leader (FTPL): we don’t think it works directly, because FTPL requires the reward function to be concave, but our leader’s utility function is not concave (not even continuous) in her strategy x.  Other ideas are needed to design algorithms for the discontinuous utility function.  For example, Bernasconi et al (2023)’s reduction provides an algorithm with $O(K^{3n/2}\sqrt{T})$ regret under action feedback.  We conjecture that the $O(K^{3n/2})$ term might be improvable by the following bi-level algorithm:
>
> - Regard each best-response region as an arm in an adversarial multi-armed bandit problem, and use algorithms such as MWU and EXP3 to pick a region at each round;
>
> - Run an adversarial online linear optimization algorithm (such as FTPL in the full-info setting) inside each best-response region to pick the leader’s strategy.
>
> This approach generalizes our Algorithm 3 “UCB on best response regions”. We leave the formal analysis as future work, because we feel that the analysis for adversarial setting is beyond the scope of our current paper, which is intended for the independently interesting stochastic setting. We included this discussion in the revised PDF.
>
>
> > (Question 1) Is the improvement from $K^{3n/2}$ to $K^n$ only due to stochastic types?
>
> The improvement is not only because followers have stochastic types rather than adversarial types, but requires new algorithm designs.  Directly applying previous adversarial algorithm (Bernasconi et al, 2023) to the stochastic setting does not give $K^n \sqrt{T}$ regret.  We make use of their reduction to linear bandit problems but change their adversarial linear bandit algorithm to a stochastic linear bandit algorithm.  That allows us to reduce the regret from $K^{3n/2}\sqrt{T}$ to $K^n\sqrt{T}$.

---

### Official Review · Reviewer_Sbtn · 2025-11-01

**Soundness:** 4
**Presentation:** 3
**Contribution:** 3
**Rating:** 6
**Confidence:** 3

**Summary:**

The paper studies the online learning version of the multi-follower Bayesian Stackelberg game. The authors consider the setting where the leader knows each follower's utility function but not their private types. The paper designs learning algorithms for the leader under both type-feedback and action-feedback settings, and provides detailed regret analysis.

**Strengths:**

The paper is well-written and easy to follow. The authors conduct an in-depth analysis of the geometric properties of the game, which provides insights for the learning algorithms. I find the proposed algorithm interesting as it significantly reduce regret. The theoretic results look solid and strong to me (did not check all the proof details though). The multi-follower setting is much more general than the standard version, and the propose method can potentially improve the applicability of the Bayesian Stackelberg game.

**Weaknesses:**

There are some minor issues with the paper. The definition of $W$  in Subsection 3.2 seems inconsistent. If $W$ is a matrix, then the $i$-th element of $W$ should be $w_i: \Theta^K \mapsto A^K$, not $w_i: \Theta \mapsto A$. Am I missing something here?

**Questions:**

Please see my comments above.

---

> ### Author Response · Authors · 2025-11-23
>
> Thank you for your time and review! We are glad that you found our model to be more general than prior works and the results to be interesting and insightful!
>
> We clarify the definition of $W$ in Subsection 3.2.  It is a tuple of $n$ mappings $W = (w_1, … w_n)$ where each $w_i : \Theta \to A$ is a mapping that maps the type $\theta_i$ of follower $i$ to the action $a_i = w_i(\theta_i)$ that the follower should take.  In matrix form, each $w_i$ is a row vector $w_i = ( w_i(1), \ldots, w_i(k) )$, where the $k$-component/column $w_i(k)$ is the action that follower $i$ should take if the follower has type $\theta_i = k$.  The entire $W$ is a mapping from joint type space $\Theta^n$ to joint action space $A^n$.  We have clarified this more in the updated PDF. Please let us know if more clarifications are needed.

---

### Official Review · Reviewer_jBAa · 2025-11-03

**Soundness:** 4
**Presentation:** 4
**Contribution:** 4
**Rating:** 10
**Confidence:** 4

**Summary:**

This paper studies the problem of learning an optimal commitment against multiple followers, each of whose type is drawn from a distribution. In doing so, this paper generalizes the usual Stackelberg paradigm of optimal commitment against a single follower and even the Bayesian Stackelberg setting where the single follower’s type is drawn from a distribution.

The latter problem is known to be NP-hard in general, implying hardness for this more general problem. A natural algorithmic approach for this problem is to partition the leader’s action space into regions that correspond to best-response regions for the different follower profiles. However, the bottleneck to such an approach is that the number of partitions grows exponentially in the number of types for the follower — this problem doubly applies with multiple followers, since the number of best-response (BR) profiles has exponential dependence on the number of followers as well.

This paper surmounts this technical difficulty by using results from computational geometry to bound the number of non-empty regions with exponential dependence only on the number of “pure” actions of the learner. The paper also provides an explicit enumeration of these non-empty regions with the same asymptotic parameters, thus providing an offline algorithm for the problem. Then, they strengthen this to an online learning algorithm, which is able to learn the optimal policy at a rate faster than just learning the distributions based on intricate technical analysis. Their results cover both the settings where the type of the followers is seen at the end of each round versus just their action, and provide lower bounds with the same exponential dependence.

**Strengths:**

The paper provides a novel technique that breaks through the barrier of exponential best response profiles with multiple followers/ follower types, by focusing on the dimension of the leader's action space. This allows for a fine-grained view of the complexity of finding an optimal commitment and opens the door for the upper bound results in the paper which as noted, from the first positive results for multiple followers. The paper covers a variety of settings and is overall a very strong submission with novel technical contributions.

**Weaknesses:**

NA

**Questions:**

Do the results extend to settings where the action sets are arbitrary convex sets of a specified dimension, especially for the leader?

---

> ### Author Response · Authors · 2025-11-23
>
> Thank you for your insightful review and your strong endorsement for the paper. We are delighted the work resonated with you. Please see below our response to your question:
>
> > Q: Do the results extend to settings where the action sets are arbitrary convex sets of a specified dimension, especially for the leader?
>
> This is a great observation!  **Yes**, all of our results (currently stated for the probability simplex) extend to the setting where the leader’s strategy space is an arbitrary bounded and closed convex subset of the $L$ dimensional space, as long as the leader and followers’ utility functions $u(x, a)$, $v_i(x, a)$ are linear (or affine) in $x$.  In particular, our geometric partition of the leader’s strategy space into $O(n^L K^L A^{2L})$ regions, as well as our concentration inequalities for the leader’s expected utility, continue to hold because they only need the linearity of utility functions.  So all of our conclusions generalize.  We have added Footnote 1 in the revised PDF to discuss this generalization.

---

### Meta-Review · Area_Chair_14G2 · 2026-01-05

**Summary:**

All the Reviewers are positive about this paper, thus I recommend acceptance. Some Reviewers raised the concern that the paper is not completely clear in some parts. I ask the Authors to improve those parts following Reviewers' suggestions.

**Reviewer Concerns:**

The Reviewers only had minor concerns about this paper, and they have all been addressed by the Authors' rebuttals.

**Reviewer Scores:**

Reviewer jBAa, Score 10 - I believe that the rebuttal would not have changed the reviewer’s opinion, especially given the other reviews.

Reviewer Sbtn, Score: 6 - I believe that the rebuttal would not have changed the reviewer’s opinion, especially given the other reviews.

Reviewer F5NQ, Score: 6 - I believe that the rebuttal would not have changed the reviewer’s opinion, especially given the other reviews.

Reviewer av7X, Score: 6 - I believe that the rebuttal would not have changed the reviewer’s opinion, especially given the other reviews.

---

### Decision · Program_Chairs · 2026-01-26

Accept (Poster)